# Meta-Learning an In-Context Transformer Model of Human Higher Visual Cortex

**Muquan Yu**[1,2]    **Mu Nan**[1]    **Hossein Adeli**[3]    **Jacob S. Prince**[4]    **John A. Pyles**[5]

**Leila Wehbe**[6]    **Margaret M. Henderson**[6]    **Michael J. Tarr**[6]    **Andrew F. Luo**[†1]

[1] University of Hong Kong    [2] Chinese University of Hong Kong    [3] Columbia University
[4] Harvard University    [5] University of Washington    [6] Carnegie Mellon University

mqyu@link.cuhk.edu.hk    [†]Corresponding author: aluo@hku.hk

## Abstract

Understanding functional representations within higher visual cortex is a fundamental question in computational neuroscience. While artificial neural networks pretrained on large-scale datasets exhibit striking representational alignment with human neural responses, learning image-computable models of visual cortex relies on individual-level, large-scale fMRI datasets. The necessity for expensive, time-intensive, and often impractical data acquisition limits the generalizability of encoders to new subjects and stimuli. **BraInCoRL** uses in-context learning to predict voxelwise neural responses from few-shot examples *without any additional finetuning* for novel subjects and stimuli. We leverage a transformer architecture that can flexibly condition on a variable number of in-context image stimuli, learning an inductive bias over multiple subjects. During training, we explicitly optimize the model for in-context learning. By jointly conditioning on image features and voxel activations, our model learns to directly generate better performing voxelwise models of higher visual cortex. We demonstrate that BraInCoRL consistently outperforms existing voxelwise encoder designs in a low-data regime when evaluated on entirely novel images, while also exhibiting strong test-time scaling behavior. The model also generalizes to an entirely new visual fMRI dataset, which uses different subjects and fMRI data acquisition parameters. Further, BraInCoRL facilitates better interpretability of neural signals in higher visual cortex by attending to semantically relevant stimuli. Finally, we show that our framework enables interpretable mappings from natural language queries to voxel selectivity. Our code and model weights are publicly available at https://github.com/leomqyu/BraInCoRL.

## 1  Introduction

Human visual cortex transforms raw sensory input into behaviorally-relevant representations of the world. While early visual areas are characterized by retinotopic organization and selective tuning to simple features such as edges and orientation gradients [1, 2, 3, 4, 5], higher-order visual areas demonstrate selectivity to more abstract semantics and categories. While this functional organization is largely consistent across individuals at a coarse scale, the spatial distribution and fine-grained semantic selectivity within visual cortex varies due to structural differences, developmental experience, and life-long learning [6, 7, 8, 9, 10, 11]. Such functional inter-subject differences pose a fundamental challenge in constructing generalizable models of higher visual cortex that can adapt to subject-specific neural organization without exhaustive data collection for every individual.

39th Conference on Neural Information Processing Systems (NeurIPS 2025).

Recent advances in deep learning offer a promising avenue for addressing this challenge. Vision models pretrained on large-scale image datasets not only achieve strong object recognition performance, but also recapitulate hierarchical processing patterns observed in biological vision [12, 13, 14]. While these models may encapsulate some *universal principles of visual processing* [15], they do not inherently account for *individual* differences in cortical organization. To close the gap between artificial and biological systems, researchers have developed image-computable fMRI encoders – models that *predict brain activity from visual stimuli* [16]. These encoders typically regress image features onto voxelwise brain responses using subject-specific data, acting as computational probes of visual processing. Unfortunately, current approaches require many hours of costly fMRI scans per subject to fit these mappings – a prohibitive bottleneck for scalability to new populations, stimuli, and tasks, especially in clinical settings where collecting large amounts of data is difficult.

We bridge this gap with **BraInCoRL** (**Bra**in **In**-**Co**ntext **R**epresentation **L**earning), a transformer-based framework that meta-learns to predict subject-specific neural responses from provided examples. Inspired by language models that adapt to new tasks in-context, our approach treats voxel encoding as a function inference problem: given a handful of stimulus-response pairs from a new individual and novel stimuli, BraInCoRL constructs a voxelwise encoding model without any further training. By jointly optimizing for in-context learning across diverse subjects and stimuli, our model discovers shared functional principles of higher visual cortex that generalize to new subjects and stimuli represented by only a small amount of data. We illustrate the problem we are tackling in Figure 1.

We demonstrate that BraInCoRL: **(1)** Outperforms existing voxelwise encoder models in the low-data regime on novel visual stimuli while exhibiting strong generalization with increasing context. **(2)** Can generalize to new experiments with different scanning parameters. **(3)** Through analysis of attention values, learns to rely on images that are reflective of the category selected for in each region. **(4)** When paired with features from contrastive image-language models, facilitates zero-shot natural language–based characterization of cortical selectivity, enabling interpretable, finer-grained query-driven functional mapping.

## 2 Related work

**Computational Encoding and Decoding Models for Visual Cortex.** Computational modeling of neural data often involves two complementary approaches: encoding models that map from stimuli to neural activations, and decoding models that map from neural data to stimuli [16, 17, 18, 19, 20, 21, 22, 23, 24]. The development of both approaches has been facilitated by advances in machine learning models. For encoding models, the dominant approach is to combine pretrained deep feature extractors with linear voxelwise weights [25, 26, 27, 28, 29, 30]. More recent approaches have proposed to leverage transformers [31, 32] to learn the relationship between brain regions of a single subject. Most similar to our framework is the pioneering work by Adeli et al. [31] and Beliy & Wasserman et al. [33] which uses an auto-decoder based transformer network for multi-subject voxelwise encoding; However these approaches still require fine-tuning for novel subjects. More generally, encoders have been used to investigate the coding properties in higher-order visual areas [34, 35, 36, 37, 38, 39, 40, 41]. Encoders have been further combined with image-generation models [42, 43, 44, 45, 46, 47, 48, 49] or language-generation models [50, 51] to explore semantic selectivity. Recent progress on large generative models has enabled stimulus reconstruction from fMRI, EEG, and MEG signals for images [52, 53, 54, 55, 56, 57, 58, 59, 60, 61, 62, 63], videos [64, 65, 66, 67, 68, 69, 70], and language/audio [71, 72, 73, 74, 75, 76, 77].

**Representational Organization of Visual Cortex.** Human visual cortex exhibits a hierarchical organization from primary visual to higher-order visual areas. The higher visual cortex is characterized by a tiling of semantically specialization. Approaches using functional localizers have identified category selective regions in higher visual that are responsive to faces [78, 79, 80, 81], places [82, 83], bodies [84], objects [85, 86], food [87, 88, 89], and words [90, 91]. While the spatial location of these broad category-selective regions are generally consistent across subjects [92], significant inter-individual variability exists in their anatomical location, spatial extent, and response profiles [6, 7, 8, 9, 93, 94, 95, 96, 97]. Accurately characterizing visual processing in higher-order visual areas necessitates subject-specific encoding models that capture individual diversity [98].

**Meta-Learning and In-Context Learning.** Our work builds upon meta-learning and in-context learning (ICL). Meta-learning trains models to "learn to learn" from a distribution of tasks, enabling

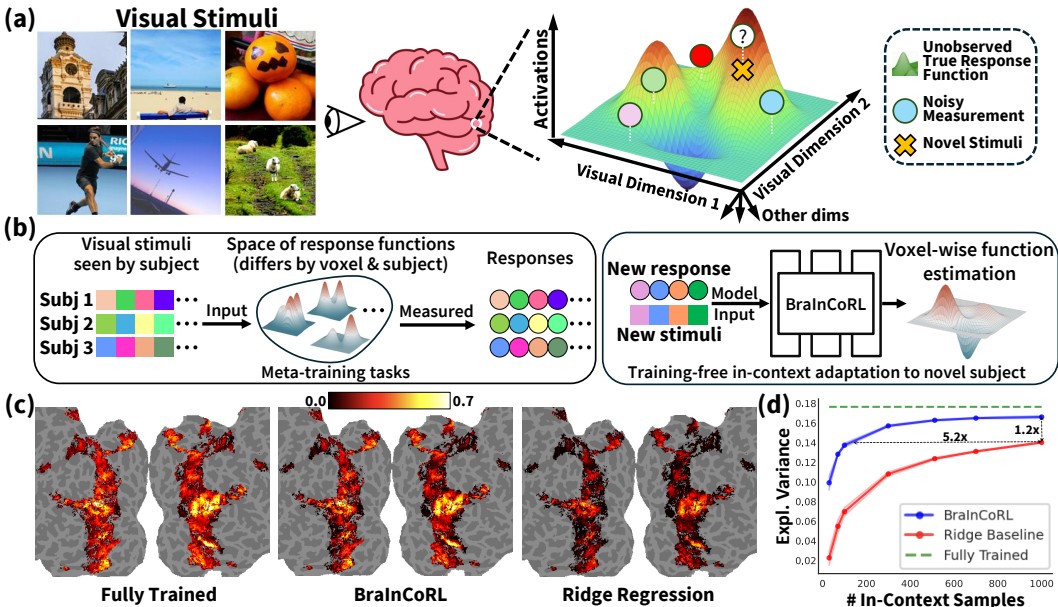

Figure 1: **BraInCoRL: Meta-Learning an In-Context Visual Cortex Encoder. (a)** The voxelwise brain encoding problem. For each voxel, there is a response function that maps from visual stimuli to voxel activation. In practice, we can only observe the noisy measurements from fMRI. The goal is to infer an image-computable function for each voxel to predict its activation. **(b)** BraInCoRL treats each voxel as a meta-learning task, and samples (image, response) pairs from multiple subjects. During testing, the model is conditioned on a small number of novel images and measurements from a new subject and directly outputs the function parameters. **(c)** From left to right, the explained variance from the full model trained on 9,000 images from one subject, BraInCoRL with only 100 in-context images from the new subject, and a baseline ridge regression also with 100 images (for this baseline, voxelwise regularization is determined using 5-way cross-validation). Our method achieves much higher data efficiency than baseline. **(d)** Explained variance as a function of in-context support set size. As the in-context support set size increases from 0 to 1,000, BraInCoRL steadily improves and approaches the fully trained reference model fit to converge on each subject's full 9,000-image training set, demonstrating high prediction accuracy and data efficiency.

quick adaptation to new tasks with few examples, via methods like meta-optimization [99, 100, 101] or metric-based approaches [102]. More recently, ICL has emerged as a powerful capability in Large Language Models [103, 104], where models adapt to new tasks at inference time solely based on examples provided in their prompt without any parameter updates [105, 106]. This has led to hypotheses that ICL is an emergent form of implicit meta-learning, where transformers effectively learn underlying learning algorithms during pre-training [107, 108]. Our goal is to learn the structure of functions that map between visual stimuli and voxelwise brain response. Our framework combines meta-training (across voxels and subjects) and in-context learning (across stimuli) to enable training free adaptation to novel subjects.

# 3 Methods

Our proposed framework leverages meta-learning and uses few-shot, in-context examples for voxelwise prediction of unseen stimuli (Figure 2). Critically, for unseen subjects, this approach does not require *any* additional finetuning. We achieve this by treating the mapping function from visual stimuli and the response of individual voxels as a set of meta-training tasks. This voxelwise approach is in line with higher-order visual areas being described by a multitude of functionally diverse voxels, which we randomly sample during training.

## 3.1 Motivation and Problem Definition

There is substantial inter-subject anatomical and functional variability in the higher visual cortex among humans. Consequently, while one can learn per-subject image-computable encoders that map image features to brain responses with high predictive accuracy, these models require large amounts

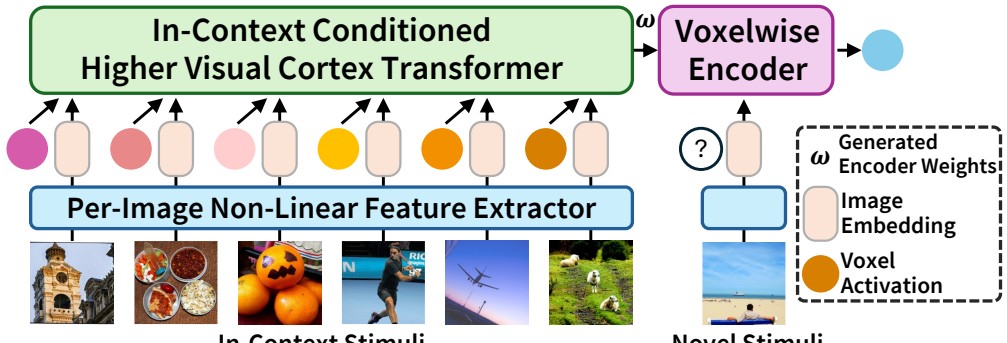

Figure 2: **Architecture of the In-Context Voxelwise Encoder (BraInCoRL). (1)** A pretrained feature extractor converts visual stimuli into vector embeddings. **(2)** A higher visual cortex transformer integrates these embeddings with voxel activations to learn context-specific features and generates hyperweights for a subsequent voxelwise encoder backbone. **(3)** The voxelwise encoder, conditioned on the hyperweights, predicts voxel responses for novel stimuli.

of within-subject data and do not exploit information across subjects. To account for this variability across individuals, we design our framework to treat individual voxels as the fundamental unit of modeling. Importantly, our method does not assume any overlap in stimuli across subjects, yet still enables us to take advantage of multi-subject training data.

We formalize this problem by assuming an image $j$ is represented as RGB tensor $\mathcal{I}_j \in \mathbb{R}^{H \times W \times 3}$. Given an image $j$ and a human subject $k$, there is a 1D array of voxel activations (beta values) from higher visual cortex: $(\beta_1, \beta_2, ..., \beta_{N_k})_j = B_{j,k} \in \mathbb{R}^{1 \times N_k}$, where the number of voxels $N$ will differ between subjects.

Given a new subject not seen during training, we have a small set of $n$ seen images $(\mathcal{I}_1, \mathcal{I}_2, ..., \mathcal{I}_n)$ and measured brain responses $(B_1, B_2, ..., B_n)$ for this new subject. *Our goal is to estimate the brain response to an arbitrary new image $\mathcal{I}_{\text{novel}}$.*

## 3.2 Meta-Learning an In-Context Transformer

Image-computable encoders that map from images to brain responses for a single subject $k$ are typically considered as a function $f_k(\mathcal{I}) \Rightarrow B$, and jointly model the entire visual cortex. While powerful, this approach cannot be easily extended to the multi-subject scenario, where test-time individuals may have functional and anatomical differences that are not known during training. In contrast, BraInCoRL considers each voxel $v$ to have a unique and unobserved visual response function $f_{k,v}(\mathcal{I}) \Rightarrow \beta_v$. Voxels can be from many different subjects. During training, we consider each voxel's response function to be a meta-training task, where each task is effectively specified by input images and voxel response pairs. In order to facilitate training-free adaptation on new subjects, we utilize in-context learning across stimuli enabled by a transformer backbone.

For a single voxel we define a support set of $p$ images and neural responses $\{(x_1, \beta_1), (x_2, \beta_2), ..., (x_p, \beta_p)\}$, where $x_i \in \mathbb{R}^m$ is the image embedding vector extracted by a frozen image feature extractor $\phi(\cdot)$, i.e., $x_i = \phi(I_i)$, and $\beta_i \in \mathbb{R}$ is the voxel's response observed for image $I_i$. Each pair is concatenated to form context tokens $c_i = [x_i; \beta_i]$, and the full context is defined as $\{c_1, ..., c_p\}$. Unlike traditional in-context inference in large language models, where there is a query concatenated to the end of the context, we avoid expensive per-voxel inference by directly generating the parameters for the voxelwise visual response function. During training, we optimize the BraInCoRL transformer $T$ with parameters $\theta$ such that it outputs voxel response function $f$ with parameters $\omega$:

$$\omega = T_\theta(c_1, c_2, ..., c_p) \tag{1}$$

$$\hat{\beta} = f_\omega(\mathcal{I}) \tag{2}$$

Since $T$ and $f$ are differentiable, we optimize $\theta$ to maximize the likelihood of observing $\beta$ given $\mathcal{I}$:

$$\theta^* = \arg\min_\theta \mathbb{E}_{(I_q, \beta_q)} \| f_\omega(\mathcal{I}) - \beta_{\text{True}} \|_2^2 \tag{3}$$

In practice, we use mean-squared-error and gradient based mini-batch optimization.

Table 1: **Voxelwise performance across five category-selective regions.** Explained variance is shown for our in-context model ("BraInCoRL") that uses just 100 in-context images, the fully trained reference model fit to converge on each subject's full 9,000-image training set ("Fully Trained"), and within-subject ridge regression baselines (100, 300 within-subject test images), plus the FsAverage map averages over other subjects. Our model outperforms both subject-wise and anatomical baselines, and demonstrates strong data-efficiency.

| | Faces | | Places | | Bodies | | Words | | Food | | Mean | |
|---|---|---|---|---|---|---|---|---|---|---|---|---|
| | S1 | S2 | S1 | S2 | S1 | S2 | S1 | S2 | S1 | S2 | S1 | S2 |
| Fully Trained | 0.19 | 0.16 | 0.20 | 0.27 | 0.28 | 0.24 | 0.11 | 0.11 | 0.16 | 0.17 | 0.18 | 0.19 |
| Ridge-100 | 0.10 | 0.07 | 0.08 | 0.14 | 0.16 | 0.12 | 0.02 | 0.03 | 0.05 | 0.07 | 0.07 | 0.08 |
| Ridge-300 | 0.13 | 0.10 | 0.13 | 0.20 | 0.22 | 0.16 | 0.06 | 0.06 | 0.10 | 0.11 | 0.11 | 0.12 |
| FsAverage map | 0.13 | 0.06 | 0.11 | 0.19 | 0.09 | 0.08 | 0.06 | 0.03 | 0.14 | 0.18 | 0.08 | 0.06 |
| **BraInCoRL-100** | **0.16** | **0.13** | **0.16** | **0.23** | **0.25** | **0.21** | **0.07** | **0.08** | **0.12** | **0.13** | **0.13** | **0.15** |

### 3.3 Test-time Context Scaling

At test time, when we encounter a new subject, we assume we have access to a small set of novel images and the corresponding brain responses – we want to predict a voxelwise encoder. While our voxelwise parameterization successfully resolves the challenge of unknown test-time anatomy and geometry, the challenge of unknown test-time context size remains. Unlike transformers in language models, where the output is dependent on the order of the samples, we want our model to be explicitly invariant to input order. In order to facilitate variable length context, we utilize logit scaling [109, 110, 111]. Assuming a query/key $(q, k)$ with $d_k$ features and a length $l$ context:

$$\alpha_{\text{orig}} = \frac{q \cdot k}{\sqrt{d_k}}; \quad \alpha_{\text{scaled}} = \frac{\log(l) \cdot q \cdot k}{\sqrt{d_k}} \tag{4}$$

We find this method effectively enables context scaling when trained with variable length context. While the hypernetwork could, in principle, parameterize any class of neural encoders (e.g., MLPs, convolution, attention layers), prior studies utilizing brain data have largely used linear parameterizations that map from deep network features to voxel responses [13, 14], and find that such a choice offers high performance and interpretability. Given features $x \in \mathbb{R}^{1 \times q}$ from a pretrained neural network $x = \phi(\mathcal{I})$, we adopt the same setup and predict the final voxel response:

$$\hat{\beta} = f(\phi(\mathcal{I}); \omega) = \texttt{matmul}(x, \omega) \tag{5}$$

A more detailed description on the test-time context scaling technique is provided in Appendix A.4.

## 4 Experiments

We utilize BraInCoRL to generate encoder weights in a low-data regime. We start by describing our experiment setup. First, we evaluate the effectiveness of BraInCoRL on **novel subjects** where there is *zero overlap between the training dataset and the evaluated subject's in-context stimulus*. We also evaluate our framework where data from novel subjects are collected from a **completely different scanner and protocol**. Second, we explore the attention pattern across stimuli for different ROIs, and perform ablations to explore the need for test-time in-context stimulus diversity. Third, we show that our method enables natural language characterizations of the visual cortex using very little data.

### 4.1 Setup

**Dataset.** We primarily perform experiments with the Natural Scenes Dataset (NSD) [112], but then validate with the BOLD5000 dataset [113]. Both are large-scale neural datasets: NSD is the largest 7T fMRI image viewing dataset available, where eight subjects each viewed $\sim 10,000$ images; BOLD5000 is a 3T dataset, where four subjects each viewed $\sim 5000$ images. In NSD each image was viewed up to three times, while in BOLD5000 only a subset of images were viewed up to four times. For NSD, of the eight subjects, four subjects (S1, S2, S5, S7) completed scanning and are the focus of our analysis in the main paper. The results of other subjects are presented in the supplemental

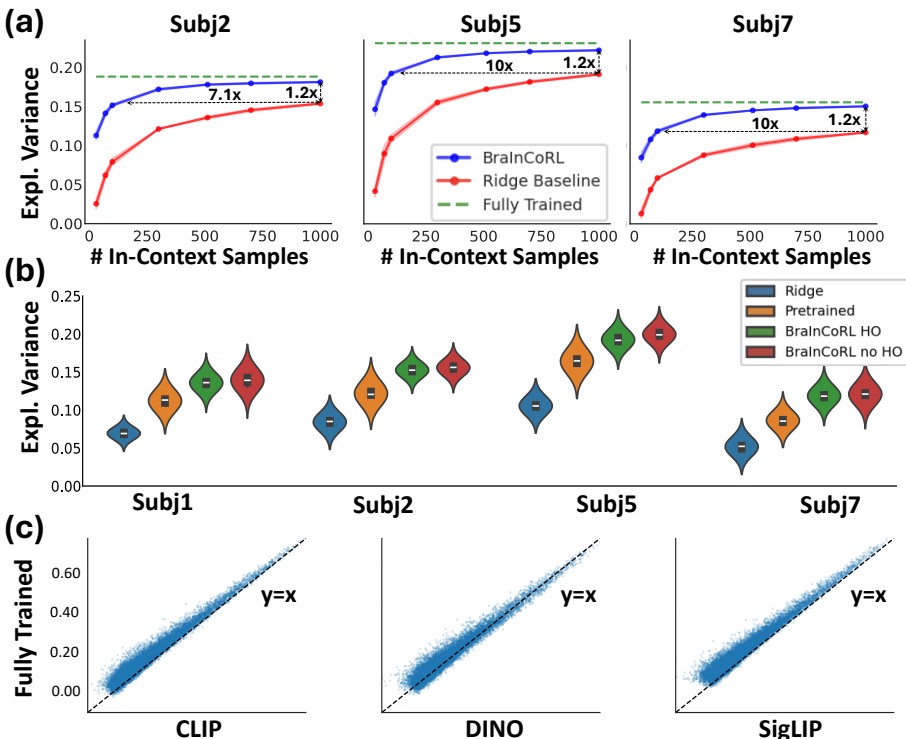

Figure 3: **Evaluation on NSD. (a)** Prediction explained variance of BraInCoRL improves on novel subjects with larger in-context support set size, outperforming within-subject ridge regression and approaching the fully trained reference model fit on each subject's full 9,000-image training set, using far less data. **(b)** Ablation (100 support images) comparing BraInCoRL variants: the original model trained while holding out the novel subject's 9,000 test-time support images ("HO"), a BraInCoRL model trained without this holdout ("no HO"), and a pretraining-only BraInCoRL model, alongside the within-subject ridge baseline. Results show that finetuning with real fMRI data improves performance, and holding out the test subject's image data does not hinder generalization. **(c)** Voxelwise explained variance from BraInCoRL (100 images) is strongly correlated with fully trained reference models across different visual encoder backbones. Note that the y-axis represents explained variance of the fully trained model (9,000 images), while x-axis represents explained variance of BraInCoRL.

materials. For each subject, $\sim 9,000$ images are unique to each other, while $\sim 1,000$ are viewed by all eight subjects. The NSD stimuli were sourced from the MS COCO dataset (as were a subset of the BOLD5000 stimuli). Unless otherwise noted, we perform our analysis in subject native volume space (func1pt8mm) for NSD, where the voxelwise betas are z-scored within each session then averaged across repeats of the same stimulus. In order to rigorously evaluate BraInCoRL for a given subject, we use the $3 \times 9,000$ unique images viewed by the other three subjects as the meta-training data. During the ROI-wise evaluation for NSD, we follow prior work [48] and apply a $t$-statistic cutoff of $t > 2$ by leveraging independent functional localizer data provided with the dataset to threshold the originally broad definitions. During quantitative evaluation, we follow prior work [114] and apply a voxel quality cutoff of ncsnr $> 0.2$. For BOLD5000, we use a model trained on four NSD subjects (S1, S2, S5, S7). Following the suggestion of BOLD5000 authors, we only model stimuli with 4 repeats and apply a cutoff of ncsnr $> 0.3$. Voxel responses are averaged across repeats.

**Training and evaluation.** Our training process takes inspiration from LLM based training setups, and we adopt a three stage process – pretraining, context extension, and supervised fine tuning. In the pretraining stage, we use an analysis-by-synthesis approach without relying on any (real) subject data. We artificially construct a large number of voxels with synthesized weights. We derive synthetic voxel responses with normally distributed noise using these synthesized weights and train our model using a fixed context size of 500. In the second stage, we randomly sample the context size from Uniform$(30, 500)$ which allows the model to acquire length robustness. Finally, in the finetuning stage, the model is trained on real fMRI data using the subject-specific beta values,

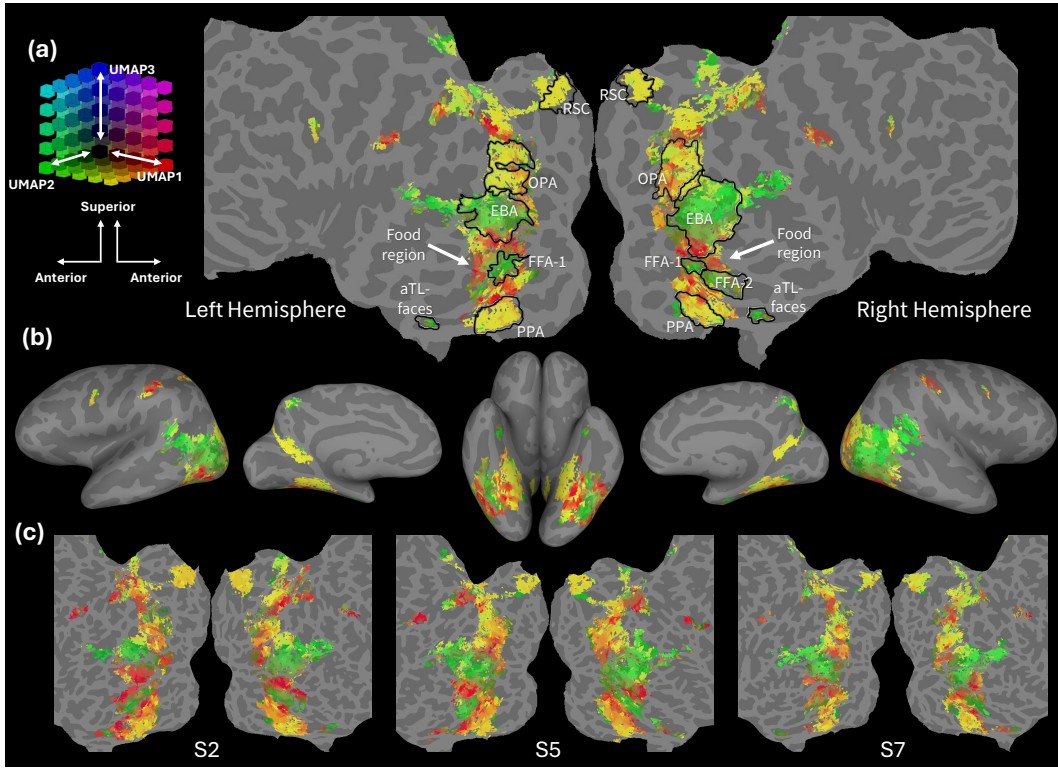

Figure 4: **UMAP visualization of predicted response weights.** We apply UMAP to BraInCoRL -predicted voxelwise weights (100 support images) and show: (a) a flatmap for S1 with ROI outlines, (b) the same projection on an inflated surface, and (c) flatmaps for S2, S5, and S7. Color-coded clusters align with body/face regions (EBA, FFA/aTL-faces), place regions (RSC, OPA, PPA), and food regions (in red).

enabling adaptation to biologically grounded neural responses. All our evaluation experiments are performed on **novel subjects** that are unseen by the model during training, with exception of (Figure 3b) no heldout ("no HO") ablation study.

## 4.2 Effectiveness of In-Context Higher Visual Cortex Encoders

**On NSD.** In this experiment, we evaluated BraInCoRL using each subject's 9,000 unique images as the in-context support set and the shared 1,000 images as the test set. For each evaluation, we randomly sampled training images from the subject-specific in-context support set and test images from the shared test set. Explained variance statistics averaged over category-selective ROIs are reported in Table 1. We compare BraInCoRL using just 100 test images against within-subject ridge regression baseline trained on 100 and 300 support-set images of the test-subject, with the regularization strength selected via 5-fold cross-validation over $[10^{-3}, 10^{-2}, \ldots, 10^{8}]$. Remarkably, BraInCoRL with only 100 images nearly matches the performance of the fully supervised reference model that is trained by gradient descent on each subject's entire within-subject support-set of 9,000 images until convergence. We also evaluate an anatomical FsAverage baseline which aligns each training subject's anatomy to a common template and projects the average response onto novel subjects for prediction. While this baseline benefits from a strong anatomical prior, it is outperformed by BraInCoRL, which directly adapts to each subject's unique neural responses with higher efficiency.

To evaluate test-time behavior, we assess how performance scales with increasing in-context support size. BraInCoRL consistently outperforms within-subject ridge regression and more efficiently approaches the fully trained reference model (Figure 1c for subject 1 and Figure 3a for subject 2, 5, 7). Moreover, we conduct ablations by evaluating a BraInCoRL model trained without holding out the test subject's support images and a BraInCoRL model with only pretraining. Results confirm

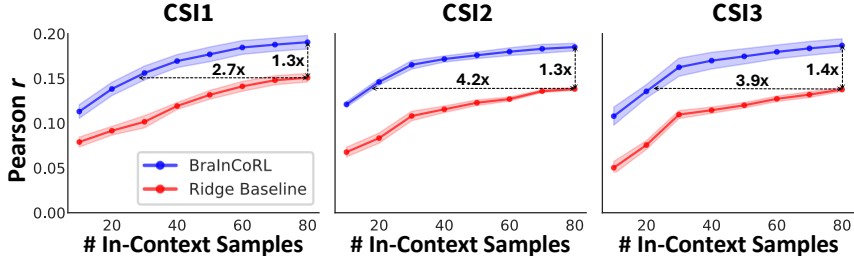

Figure 5: **Evaluation on BOLD5000.** We evaluate BraInCoRL on the BOLD5000 dataset, which was collected using a different scanner than NSD. For varying in-context support set sizes, we report voxelwise Pearson correlation between predicted and true responses for both BraInCoRL and wi̇ .

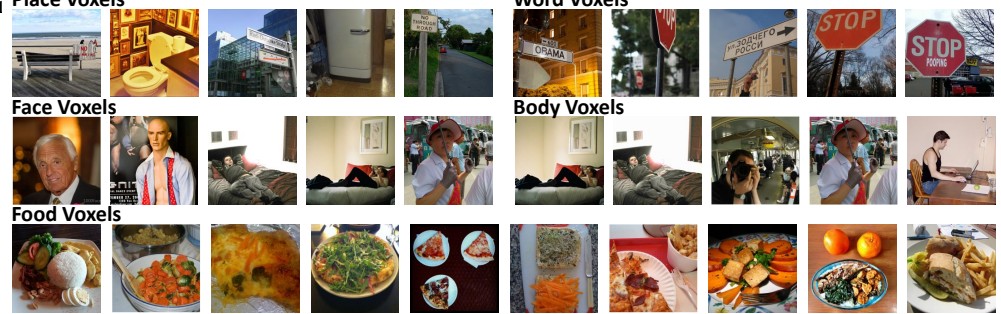

Figure 6: **Top contributing support images for each category-selective region in S1.** For each of the five category-selective regions, we select the in-context support images with the highest attention weights in BraInCoRL's final attention layer for voxels in that region. We visualize the top 5 contributing images for the place, word, face and body regions, and the top 10 for the food region.

that finetuning with real neural data boosts performance and that BraInCoRL can generalize well to previously unseen images without overfitting (Figure 3b). Additionally, we observe high voxelwise explained variance correlation between BraInCoRL and the fully trained reference model across multiple backbones (Figure 3c). Finally, we apply UMAP to the BraInCoRL predicted response-function weights, revealing clear semantic clustering across higher visual areas (Figure 4) that correspond with known visual regions.

**On BOLD5000.** We validate generalization on the BOLD5000 dataset in Figure 5. BOLD5000 has many differences with NSD and represents the challenge of cross-site generalization that is the main objective of our method. BOLD5000 was collected on a 3T scanner with a different stimulus presentation time, a slow-event related trial structure (10s inter-trial interval), different images and image datasets, a different voxel size (2mm isotropic), and different subjects. BraInCoRL achieves higher voxelwise Pearson correlations than within-subject ridge regression. Moreover, results remain consistent across different subjects, demonstrating the robustness and reliability of our method.

### 4.3 Semantic Discovery through Text–Image Embedding Alignment

To better understand how BraInCoRL leverages in-context examples, we analyze its internal attention mechanisms to identify images that strongly influence voxel predictions in category-selective regions. In Figure 6, we examine attention weights from BraInCoRL's final attention layer to determine the top-contributing images for each cortical region. The visualized images with the highest attention scores closely align with known semantic preferences of the respective cortical regions.

However, Figure 7 reveals a counterintuitive finding regarding the semantic specificity of in-context support sets. We systematically vary the specificity of the 100-image sets provided to the model, ranging from highly relevant to random selections. Selections are determined via the first text-prompt in each category (see Appendix). We observe that randomly selected images lead to better predictive performance compared to sets composed solely of highly relevant images. This suggests that overly specific context sets may limit the generalization capabilities of the encoder system, and diverse, less semantically constrained images provide richer context for learning robust voxel representations.

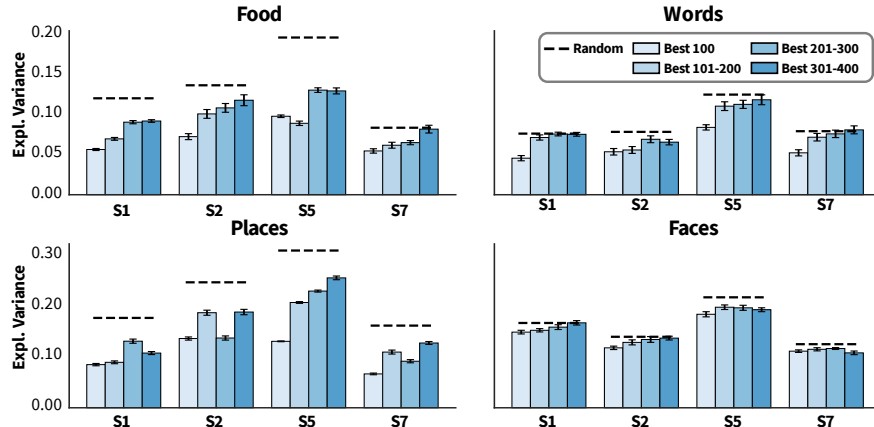

Figure 7: **Impact of support-set specificity on category-selective ROI encoding performance on NSD.** We construct in-context support sets of 100 images based on descending semantic relevance for each ROI (tiers: 1–100, 101–200, 201–300, 301–400) and compare them with randomly sampled sets of equal size. Mean explained variance in the target category-selective ROIs increases as semantic specificity decreases, with all curated sets performing worse than random sampling. This suggests that overly specific support sets hinder generalization in voxelwise encoding. This pattern echoes prior findings on diverse stimuli contributing to better encoders [13].

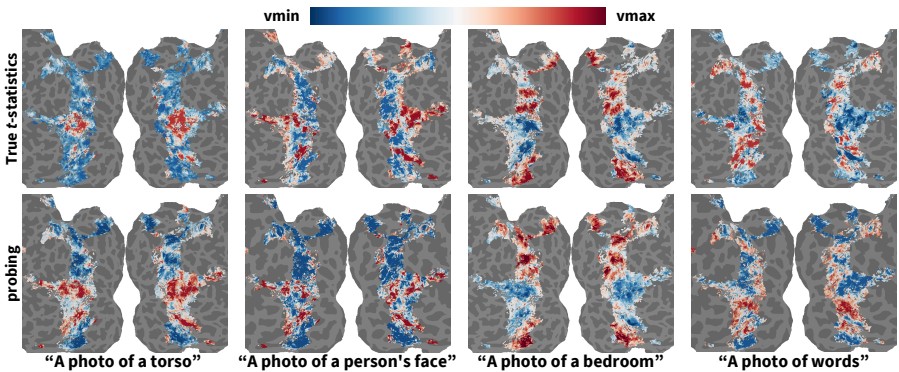

Figure 8: **Predicting cortical responses from natural language prompts.** For each semantic category, we convert a natural language prompt into a CLIP text embedding, project it into the image feature space, and use BraInCoRL to predict the corresponding voxel activation map. The predicted activations align closely with true $t$-statistic of category-selective regions (derived from fMRI functional localizer experiments), illustrating the potential for efficient, language-driven functional mapping of visual cortex.

Table 2: **Voxelwise prompt classification accuracy.** Each cell shows the percentage of voxels in a given category selective region (columns) whose peak predicted activation was elicited by a specific semantic prompt (rows, see Appendix). Using only 100 support images, BraInCoRL effectively localizes category-selective regions with high data efficiency.

|        | Bodies |      | Faces |      | Places |      | Food |      | Words |      |
|--------|--------|------|-------|------|--------|------|------|------|-------|------|
|        | S1     | S2   | S1    | S2   | S1     | S2   | S1   | S2   | S1    | S2   |
| Bodies | **0.63** | **0.54** | 0.30 | 0.16 | 0.05 | 0.03 | 0.15 | 0.19 | **0.43** | 0.17 |
| Faces  | 0.30   | 0.25 | **0.60** | **0.56** | 0.05 | 0.01 | 0.07 | 0.04 | 0.15 | 0.16 |
| Places | 0.02   | 0.09 | 0.02  | 0.05 | **0.81** | **0.88** | 0.10 | 0.07 | 0.05 | 0.10 |
| Food   | 0.04   | 0.10 | 0.08  | 0.18 | 0.08 | 0.06 | **0.66** | **0.64** | 0.31 | **0.45** |
| Words  | 0.01   | 0.03 | 0.00  | 0.04 | 0.01 | 0.02 | 0.02 | 0.05 | 0.05 | 0.12 |

### 4.4 Characterizing higher visual cortex with text embeddings to images

In this experiment, we investigate the capability of BraInCoRL to enable interpretable, query-driven functional mapping using natural language prompts.

In Figure 8, we demonstrate that natural language prompts can be effectively mapped to predicted voxel activations. For each category selective region, we convert the corresponding natural language prompt into a CLIP embedding and project it into the image feature space to directly predict voxel activations. The resulting activation maps closely match expected $t$-statistics, reflecting BraInCoRL's ability to support intuitive, language-driven cortical queries.

In the second analysis (Table 2), we quantitatively assess the accuracy of prompt-driven activation predictions. We measure the fraction of voxels within each category-selective region whose peak predicted activation aligns with the category indicated by the natural language query. Results confirm that BraInCoRL paired with language embeddings achieves a high level of alignment between predicted voxel selectivity and query semantics across multiple subjects. The predictions for word-selective voxels were notably less accurate. We hypothesize this discrepancy arises from the developmental and experiential variability inherent to the formation of word-selective regions, as these areas form predominantly through individualized learning experiences, such as reading and linguistic exposure, leading to greater inter-subject variability.

## 5 Discussion

**Limitations and Future Work.** Here we have shown that meta-learning an in-context model of higher visual cortex can yield high performance and strong data efficiency gains, outperforming anatomical (FsAverage) and subject-wise baselines on novel subjects. Our work currently focuses on static natural images, and extensions to dynamic stimuli would likely require a rethinking of the encoder backbone and network structure. Further, while we show strong generalization across scanners and utilize the largest fMRI dataset that is NSD, there may still be limitations in dataset diversity [115]. Collection of larger and more diverse fMRI datasets will help mitigate this issue.

**Conclusion.** We introduce a foundation model that serves as an fMRI encoder, mapping from natural images to voxelwise activations. We demonstrate that our method can adapt without any finetuning to new stimuli, subjects, scanners, and scanning protocols. Our model achieves this by meta-learning across voxels from different subjects, and performing in-context learning across stimuli. Our approach has significant data-efficiency and performance gains over baseline methods, and has the potential to help understand cortical structure in data-constrained environments.

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

# A  Technical Appendices and Supplementary Material

## Sections

## A.1 Social impacts

Our work introduces BraInCoRL, a meta-learning framework that uses fMRI-measured voxel responses and trained visual encoders to perform in-context adaptation: given a small support set of image–response pairs, the model directly estimates each voxel's response-function parameters for novel stimuli. BraInCoRL's fusion of diverse image data and neural measurements uncovers data-driven principles of cortical organization beyond both traditional hypothesis-driven experiments and computational encoding models that require thousands of samples. Moreover, its alignment of neural responses with natural language prompts enables the generation of new hypotheses about semantic representation in visual cortex. As such, BraInCoRL may accelerate early diagnosis and monitoring of visual or neurological disorders via rapid, subject-specific cortical mapping; guide more efficient experimental design through optimized stimulus selection; deepen our understanding of semantic coding and inter-subject variability; and, when integrated with generative image models, open avenues for brain-guided stimulus synthesis and personalized neuroprosthetic and brain–computer interface development. While BraInCoRL offers significant potential for neuroscience and clinical applications, its reliance on fMRI datasets and computational infrastructure may limit accessibility for under-resourced research groups and raise privacy concerns if applied to sensitive neural data.

## A.2 Implementation details

**Network architecture.** Our BraInCoRL model architecture comprises three main components:

1. **Input projection.** An input projection MLP layer is applied to each token individually, which maps the stimulus semantics and voxel activation into an embedding. In detail, we concatenate each image embedding with its scalar neural response and pass it through a single-layer MLP to align the two modalities into a unified internal feature space.

2. **Transformer encoder.** A stack of 20 self-attention layers integrates information across all support examples (plus learnable tokens), capturing contextual relationships and the relative importance of each in-context sample. We adopt best practices and utilize SwiGLU activation paired with pre-normalization in the attention block. We utilize 10 heads in each multi-head self attention.

3. **Weight prediction.** The [CLS] token from the final layer goes through an MLP to output a hyperweight which is used to parameterize the final encoder. In detail, the aggregated representation is fed through another single-layer MLP that outputs a weight vector, which is then used to linearly combine unknown-image embeddings to produce the final neural response predictions.

The CLIP-based variant (encoding dimension $E = 512$) contains approximately 97.2 M parameters; DINO ($E = 768$) and SIGLIP ($E = 1152$) variants comprise roughly 112 M and 130 M parameters, respectively.

**Model training.** Training is implemented in PyTorch on eight NVIDIA RTX 6000 Ada GPUs (48 GB each). We optimize using AdamW (decoupled weight decay $1 \times 10^{-4}$) with an initial learning rate of $1 \times 10^{-3}$, which decays to $1 \times 10^{-5}$ via a ReduceLROnPlateau scheduler (factor 0.1, patience 5 epochs, cooldown 2 epochs, threshold 1e-4). Mini-batches randomly sample an in-context support set of 100 images in the first pretraining stage, and randomly sample between 30 and 500 in-context support images in the second context extension stage and the third finetuning stages. Each training stage runs for up to 100 epochs with early stopping based on validation loss (patience: 5 epochs). The training batch size is 80. We allocate 20% of the test set data for validation.

**Computational cost.** With an in-context support set of 100 images, our model predicts responses for $\sim 20{,}000$ voxels in the higher visual cortex in under 20 seconds on a single RTX 6000 Ada GPU.

### A.3 Text prompts for category-selective regions

We define a set of natural language prompts for each semantic category. For every image–prompt pair, we use the CLIP text encoder to generate text encodings. The natural language prompts for each category are listed below:

**Faces** `[A photo of a person's face, A portrait photo of a face, A face facing the camera, A photo of a face, A photo of a human face, A photo of faces, People looking at the camera, A portrait of a person, A portrait photo]`

**Bodies** `[A photo of a torso, A photo of limbs, A photo of bodies, A photo of a person, A photo of people, A photo of a body, A person's arms, A person's legs, A photo of hands]`

**Places** `[A photo of a bedroom, A photo of an office, A photo of a hallway, A photo of a doorway, A photo of interior design, A photo of a building, A photo of a house, A photo of nature, A photo of a landscape]`

**Food** `[A photo of food, A photo of cuisine, A photo of fruit, A photo of foodstuffs, A photo of a meal, A photo of bread, A photo of rice, A photo of a snack, A photo of pastries]`

**Words** `[A photo of words, A photo of glyphs, A photo of a glyph, A photo of text, A photo of numbers, A photo of a letter, A photo of letters, A photo of writing, A photo of text on an object]`

## A.4 A more detailed description of logit scaling

The motivating factor underlying logit scaling is our desire to have our in-context learned encoder perform well regardless of the number of stimuli given to the model, and effectively generalize to context sizes beyond those seen during training. For example, while we may only train with between 30 and 500 images, a third-party experimenter may want to use fewer than 30 images or more than 500 images to condition the model. Across all cases, we want the model to succeed.

This logit scaling method was first proposed in [1, 2], and later adapted in Qwen LLM (logn-scaling) [3] and Llama 4 LLM (temperature scaling) [4]. We will briefly summarize the high-level math, which we take from [1] with our commentary:

Let the attention value in self-attention for a particular query token $i$ to value token $j$ to be

$$a_{i,j} = \frac{e^{\lambda \mathbf{q}_i \cdot \mathbf{k}_j}}{\sum_{j=1}^{n} e^{\lambda \mathbf{q}_i \cdot \mathbf{k}_j}}$$

Then the entropy is defined as

$$\mathcal{H}_i = -\sum_{j=1}^{n} a_{i,j} \log a_{i,j}$$

Substituting the expression for $a_{i,j}$ we have the entropy as

$$\mathcal{H}_i = -\sum_{j=1}^{n} a_{i,j} \log \left( \frac{e^{\lambda \mathbf{q}_i \cdot \mathbf{k}_j}}{\sum_{j=1}^{n} e^{\lambda \mathbf{q}_i \cdot \mathbf{k}_j}} \right)$$

Since $\sum_{j=1}^{n} a_{i,j} = 1$,

we can express the formula as

$$\mathcal{H}_i = \log \sum_{j=1}^{n} e^{\lambda \mathbf{q}_i \cdot \mathbf{k}_j} - \frac{\sum_{j=1}^{n} e^{\lambda \mathbf{q}_i \cdot \mathbf{k}_j} (\lambda \mathbf{q}_i \cdot \mathbf{k}_j)}{\sum_{j=1}^{n} e^{\lambda \mathbf{q}_i \cdot \mathbf{k}_j}}$$

If the first term is expressed as an expectation over $j$, we have

$$\sum_{j=1}^{n} e^{\lambda \mathbf{q}_i \cdot \mathbf{k}_j} = n \times \frac{1}{n} \sum_{j=1}^{n} e^{\lambda \mathbf{q}_i \cdot \mathbf{k}_j} \approx n \mathbb{E}_j [e^{\lambda \mathbf{q}_i \cdot \mathbf{k}_j}]$$

Which leads to the following approximation of entropy

$$\mathcal{H}_i \approx \log n + \log \mathbb{E}_j [e^{\lambda \mathbf{q}_i \cdot \mathbf{k}_j}] - \frac{\lambda \mathbb{E}_j [e^{\lambda \mathbf{q}_i \cdot \mathbf{k}_j} (\mathbf{q}_i \cdot \mathbf{k}_j)]}{\mathbb{E}_j [e^{\lambda \mathbf{q}_i \cdot \mathbf{k}_j}]}$$

If the vectors are assumed to be the output of a layernorm layer with length $\sqrt{d}$, the expectation can be converted to one over the angles between vectors

$$\mathcal{H}_i \approx \log n + \log \mathbb{E}_\theta [e^{\lambda d \cos \theta}] - \frac{\lambda d \mathbb{E}_\theta [e^{\lambda d \cos \theta} \cos \theta]}{\mathbb{E}_\theta [e^{\lambda d \cos \theta}]}$$

Since most randomly distributed vectors in higher dimensions are orthogonal, we derive a term which can roughly be expressed as

$$\mathcal{H}_i \approx \log n + C$$

where $C$ does not depend on the number of tokens $n$.

This leads to an approximate scaling factor for the logits of $\log n$ to keep the entropy invariant to context length.

Therefore, we change the standard formulation of attention values by applying a scaling factor of $\log n$ to the $QK^T$ term, as shown in Equation (4) of our paper.

Note that in the above explanation we adopt the notation from [1].

## A.5 Cortex prediction explained variance for different image encoding backbones and subjects

In this section, we compare three methods across multiple subjects (S1-S8) and embedding backbones (CLIP, DINO, SigLIP): the fully trained reference model fit to converge on each subject's full 9,000-image training set; our BraInCoRL approach, which adapts to a new subject with only 100 in-context images; and a within-subject ridge regression baseline trained on the same 100 images with the BraInCoRL approach. In every case, BraInCoRL outperforms ridge regression and achieves accuracy similar to that of the fully trained model.

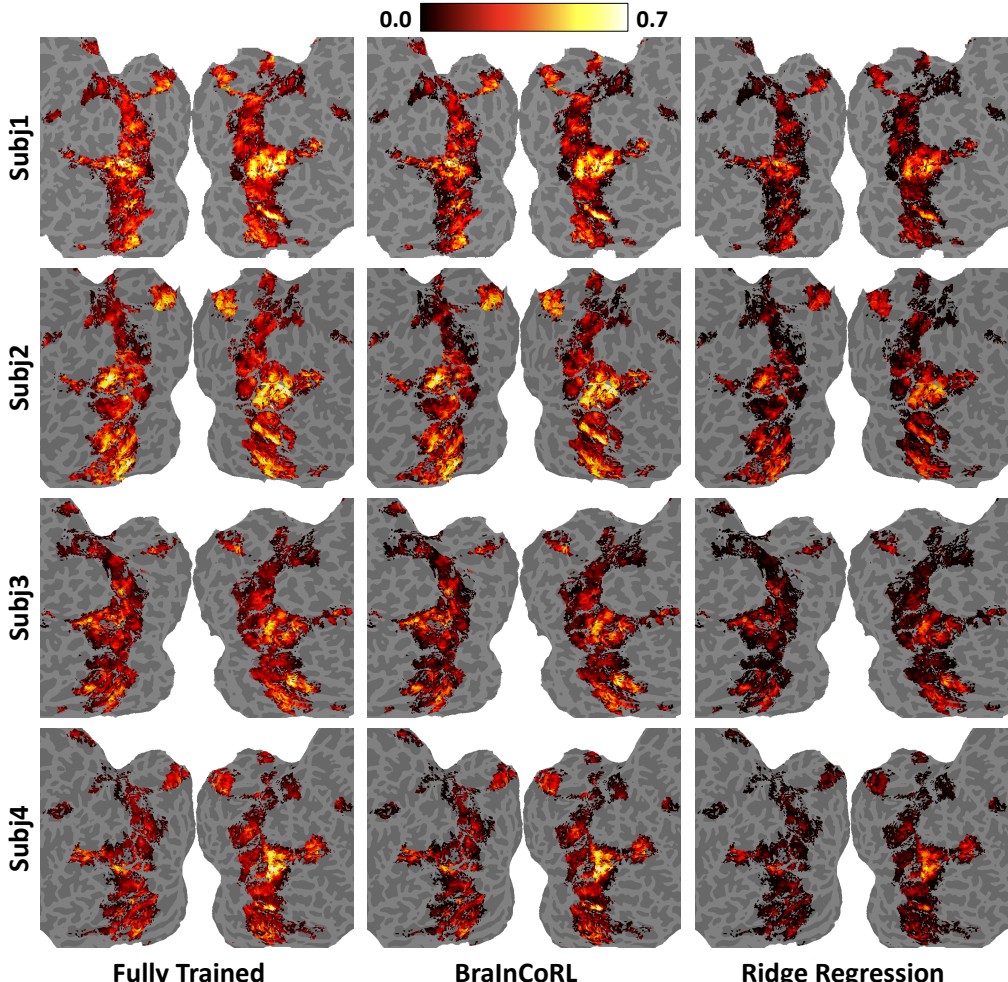

Figure S.1: **Higher visual cortex explained variance of CLIP backbone.** From left to right, we show the explained variance from the full model trained on 9000 images for a subject, BraInCoRL with just 100 in-context images from the new subject, and within-subject ridge regression with 100 images using CLIP backbone for subject 1,2,3,4.

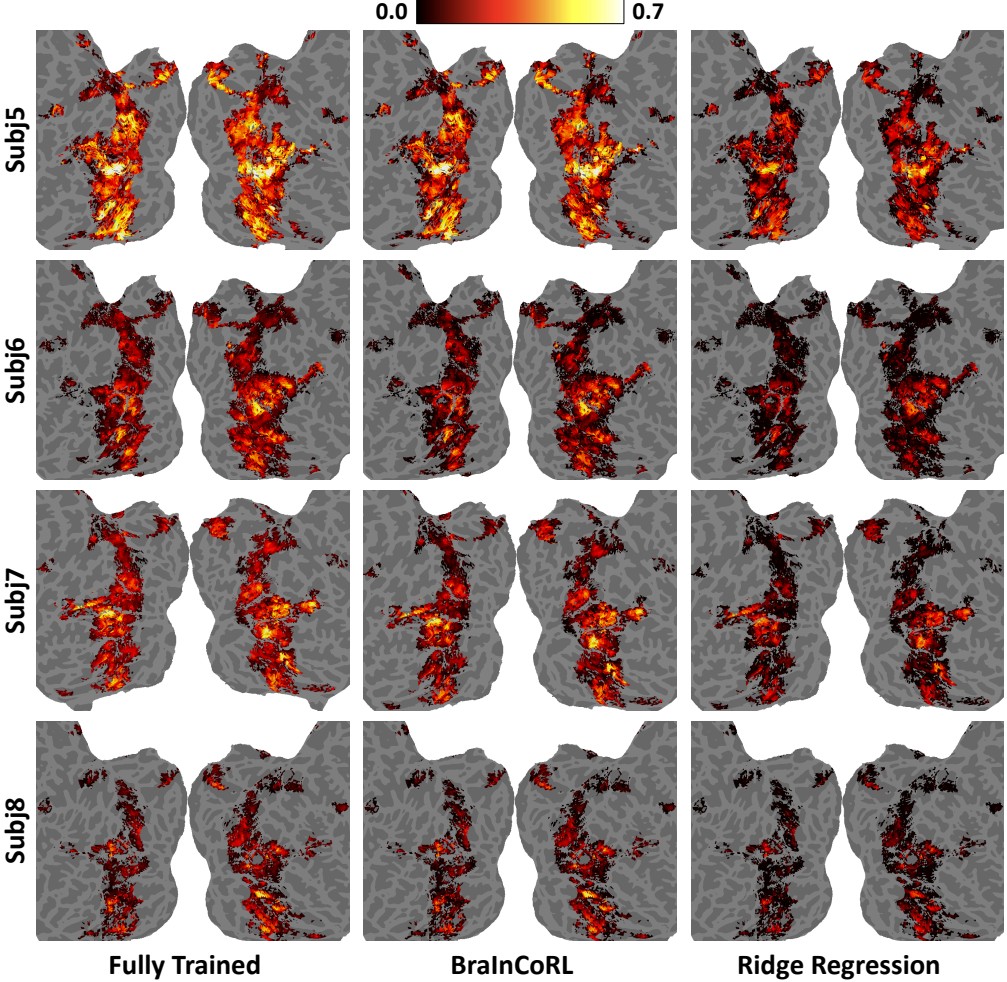

Figure S.2: **Higher visual cortex explained variance of CLIP backbone.** From left to right, we show the explained variance from the full model trained on 9000 images for a subject, BraInCoRL with just 100 in-context images from the new subject, and within-subject ridge regression with 100 images using CLIP backbone for subject 5, 6, 7, 8.

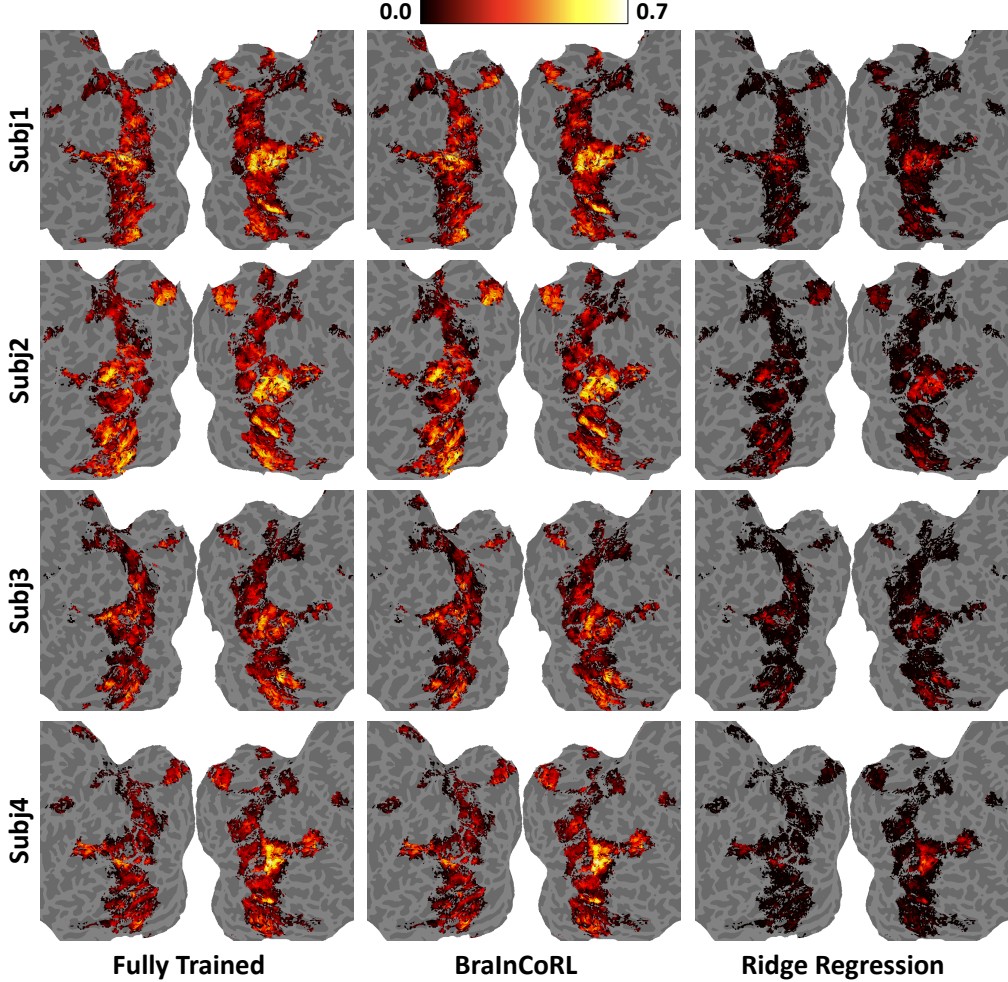

Figure S.3: **Higher visual cortex explained variance of DINO backbone.** From left to right, we show the explained variance from the full model trained on 9000 images for a subject, BraInCoRL with just 100 in-context images from the new subject, and within-subject ridge regression with 100 images using DINO backbone for subject 1, 2, 3, 4.

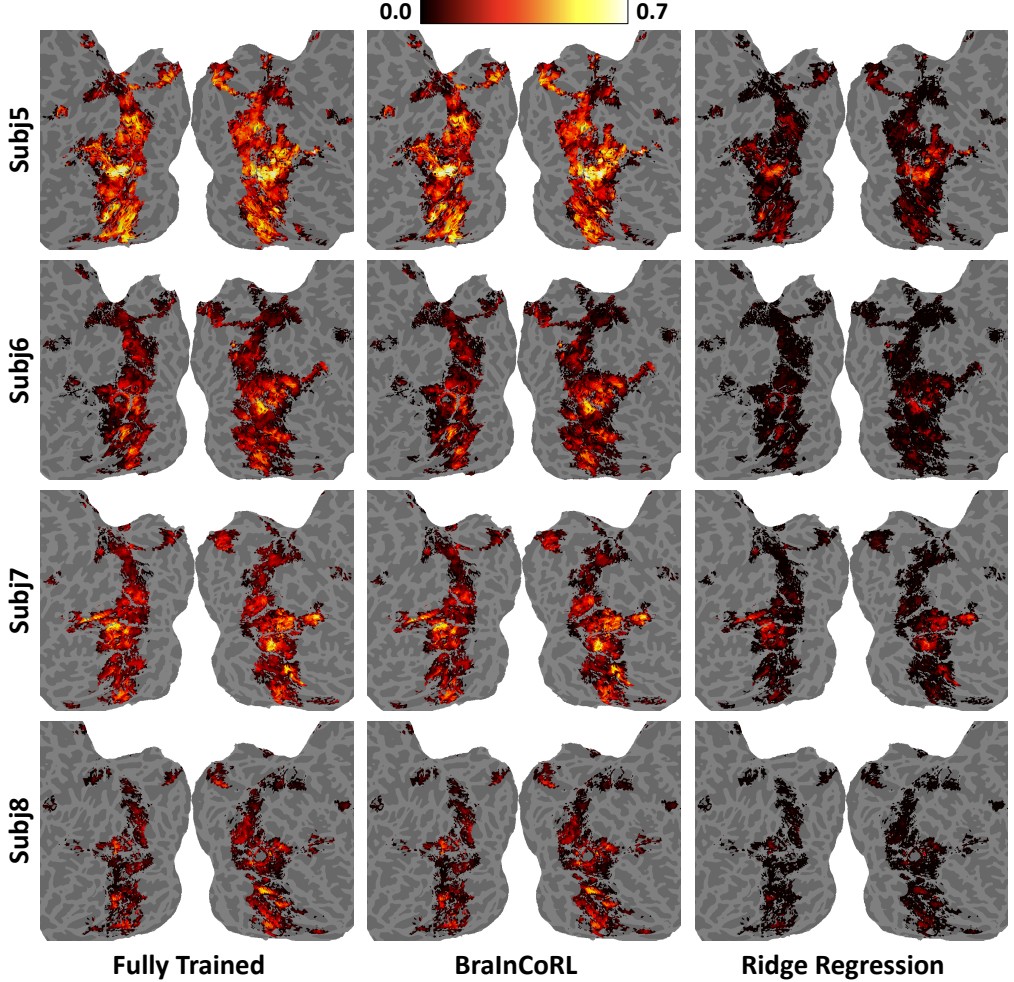

Figure S.4: **Higher visual cortex explained variance of DINO backbone.** From left to right, we show the explained variance from the full model trained on 9000 images for a subject, BraInCoRL with just 100 in-context images from the new subject, and within-subject ridge regression with 100 images using DINO backbone for subject 5, 6, 7, 8.

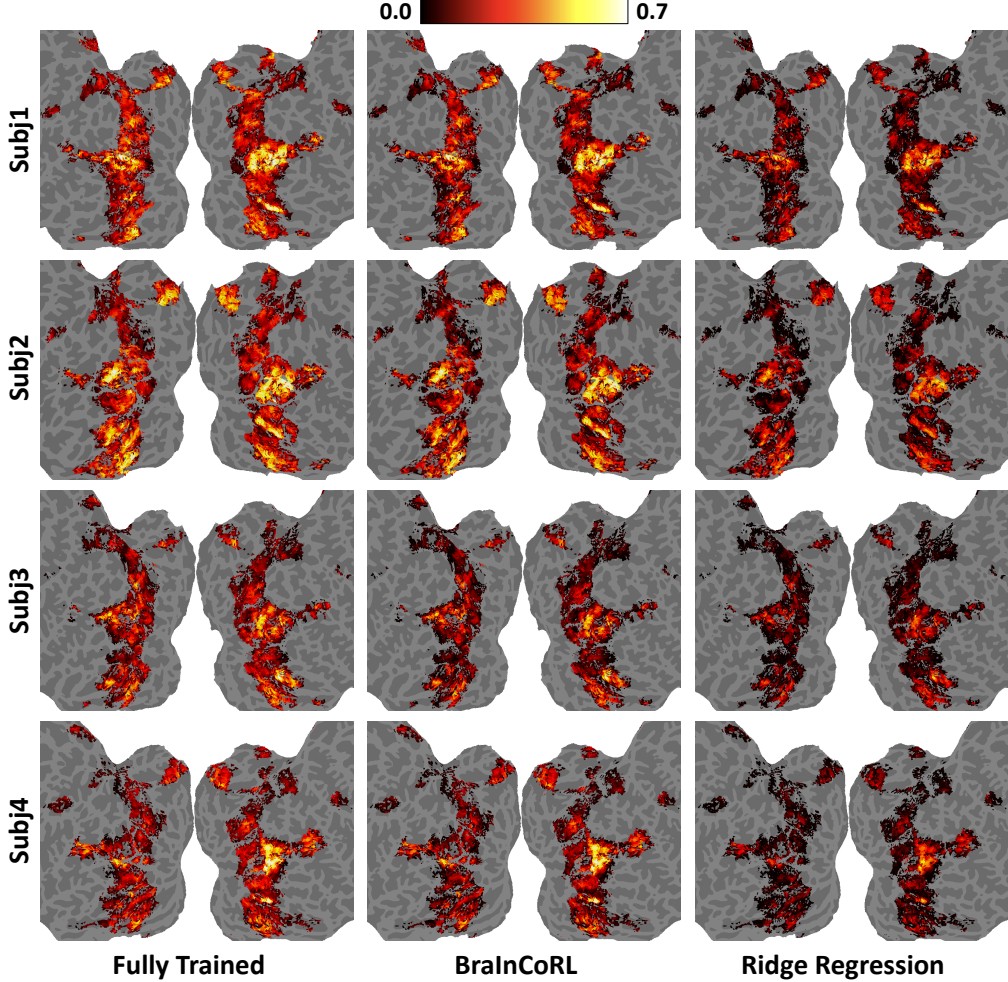

Figure S.5: **Higher visual cortex explained variance of SigLIP backbone.** From left to right, we show the explained variance from the full model trained on 9000 images for a subject, BraInCoRL with just 100 in-context images from the new subject, and within-subject ridge regression with 100 images using SigLIP backbone for subject 1, 2, 3, 4.

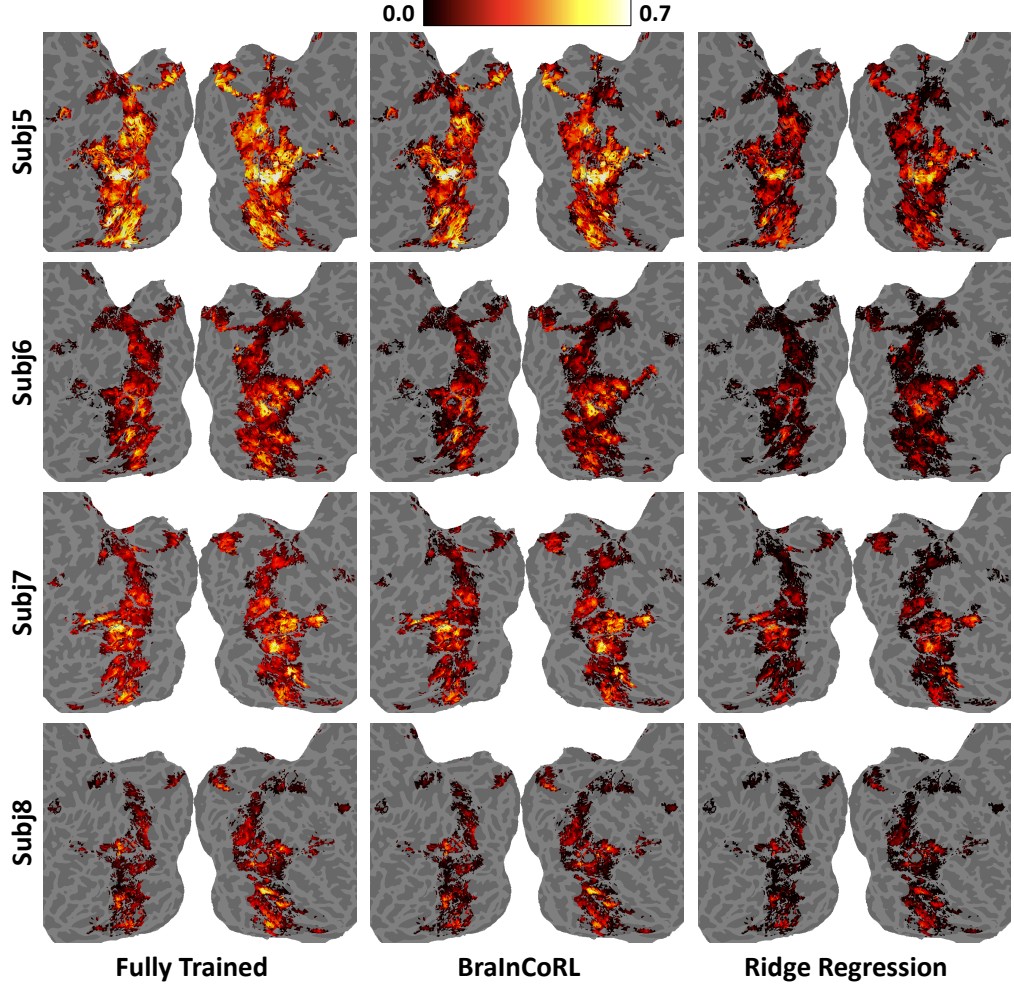

Figure S.6: **Higher visual cortex explained variance of SigLIP backbone.** From left to right, we show the explained variance from the full model trained on 9000 images for a subject, BraInCoRL with just 100 in-context images from the new subject, and within-subject ridge regression with 100 images using SigLIP backbone for subject 5, 6, 7, 8.

## A.6 Voxelwise performance across five category-selective regions for different image encoding backbones and subjects

In this section, we report voxel-wise explained variance in five category-selective regions (faces, places, bodies, words, and food) with CLIP, DINO and SigLIP backbone for subjects S1-S8. We compare our in-context model (BraInCoRL) against the fully trained reference model fit to converge on each subject's full 9,000-image training set, within-subject ridge regression baselines trained on 100 and 300 images, and the FsAverage map. BraInCoRL outperforms the ridge baselines and closely approaches the performance of the fully trained model.

Table S.1: **Voxel-wise explained variance with the CLIP backbone for Subjects 1 and 2.** We report performance for our in-context model (BraInCoRL), the fully trained reference ("Fully Trained"), within-subject ridge regression baselines (100, 300), and the FsAverage map across five category-selective regions (faces, places, bodies, words, food).

|  | Faces | | Places | | Bodies | | Words | | Food | | Mean | |
|---|---|---|---|---|---|---|---|---|---|---|---|---|
|  | S1 | S2 | S1 | S2 | S1 | S2 | S1 | S2 | S1 | S2 | S1 | S2 |
| Fully Trained | 0.19 | 0.16 | 0.20 | 0.27 | 0.28 | 0.24 | 0.11 | 0.11 | 0.16 | 0.17 | 0.18 | 0.19 |
| Ridge-100 | 0.10 | 0.07 | 0.08 | 0.14 | 0.16 | 0.12 | 0.02 | 0.03 | 0.05 | 0.07 | 0.07 | 0.08 |
| Ridge-300 | 0.13 | 0.10 | 0.13 | 0.20 | 0.22 | 0.16 | 0.06 | 0.06 | 0.10 | 0.11 | 0.11 | 0.12 |
| FsAverage map | 0.13 | 0.06 | 0.11 | 0.19 | 0.09 | 0.08 | 0.06 | 0.03 | 0.14 | 0.18 | 0.08 | 0.06 |
| **BraInCoRL-100** | **0.16** | **0.13** | **0.16** | **0.23** | **0.25** | **0.21** | **0.07** | **0.08** | **0.12** | **0.13** | **0.13** | **0.15** |

Table S.2: **Voxel-wise explained variance with the CLIP backbone for Subjects 3 and 4.** We report performance for our in-context model (BraInCoRL), the fully trained reference ("Fully Trained"), within-subject ridge regression baselines (100, 300), and the FsAverage map across five category-selective regions (faces, places, bodies, words, food).

|  | Faces | | Places | | Bodies | | Words | | Food | | Mean | |
|---|---|---|---|---|---|---|---|---|---|---|---|---|
|  | S3 | S4 | S3 | S4 | S3 | S4 | S3 | S4 | S3 | S4 | S3 | S4 |
| Fully Trained | 0.16 | 0.14 | 0.16 | 0.16 | 0.17 | 0.17 | 0.09 | 0.07 | 0.10 | 0.12 | 0.13 | 0.14 |
| Ridge-100 | 0.07 | 0.05 | 0.08 | 0.05 | 0.08 | 0.08 | 0.02 | 0.01 | 0.03 | 0.04 | 0.05 | 0.05 |
| Ridge-300 | 0.10 | 0.09 | 0.11 | 0.10 | 0.11 | 0.12 | 0.04 | 0.04 | 0.05 | 0.07 | 0.08 | 0.09 |
| FsAverage map | 0.10 | 0.03 | 0.14 | 0.05 | 0.11 | 0.06 | 0.07 | 0.03 | 0.10 | 0.07 | 0.10 | 0.04 |
| **BraInCoRL-100** | **0.12** | **0.10** | **0.13** | **0.13** | **0.14** | **0.13** | **0.05** | **0.04** | **0.07** | **0.08** | **0.10** | **0.10** |

Table S.3: **Voxel-wise explained variance with the CLIP backbone for Subjects 5 and 6.** We report performance for our in-context model (BraInCoRL), the fully trained reference ("Fully Trained"), within-subject ridge regression baselines (100, 300), and the FsAverage map across five category-selective regions (faces, places, bodies, words, food).

|  | Faces | | Places | | Bodies | | Words | | Food | | Mean | |
|---|---|---|---|---|---|---|---|---|---|---|---|---|
|  | S5 | S6 | S5 | S6 | S5 | S6 | S5 | S6 | S5 | S6 | S5 | S6 |
| Fully Trained | 0.24 | 0.17 | 0.32 | 0.13 | 0.26 | 0.18 | 0.17 | 0.09 | 0.24 | 0.09 | 0.23 | 0.11 |
| Ridge-100 | 0.11 | 0.07 | 0.16 | 0.03 | 0.13 | 0.09 | 0.06 | 0.01 | 0.11 | 0.02 | 0.10 | 0.03 |
| Ridge-300 | 0.16 | 0.11 | 0.24 | 0.07 | 0.19 | 0.13 | 0.10 | 0.04 | 0.16 | 0.04 | 0.15 | 0.06 |
| FsAverage map | 0.07 | 0.05 | 0.11 | 0.08 | 0.06 | 0.04 | 0.05 | 0.06 | 0.08 | 0.04 | 0.07 | 0.05 |
| **BraInCoRL-100** | **0.20** | **0.14** | **0.29** | **0.10** | **0.23** | **0.15** | **0.12** | **0.05** | **0.19** | **0.05** | **0.19** | **0.08** |

Table S.4: **Voxel-wise explained variance with the CLIP backbone for Subjects 7 and 8.** We report performance for our in-context model (BraInCoRL), the fully trained oracle ("Fully Trained"), within-subject ridge regression baselines (100, 300), and the FsAverage map across five category-selective regions (faces, places, bodies, words, food).

| | Faces | | Places | | Bodies | | Words | | Food | | Mean | |
|---|---|---|---|---|---|---|---|---|---|---|---|---|
| | **S7** | **S8** | **S7** | **S8** | **S7** | **S8** | **S7** | **S8** | **S7** | **S8** | **S7** | **S8** |
| Fully Trained | 0.14 | 0.08 | 0.18 | 0.10 | 0.21 | 0.09 | 0.12 | 0.04 | 0.12 | 0.06 | 0.16 | 0.09 |
| Ridge-100 | 0.06 | 0.03 | 0.08 | 0.04 | 0.11 | 0.04 | 0.03 | 0.00 | 0.03 | 0.02 | 0.06 | 0.03 |
| Ridge-300 | 0.09 | 0.05 | 0.12 | 0.06 | 0.15 | 0.05 | 0.06 | 0.01 | 0.06 | 0.03 | 0.09 | 0.05 |
| FsAverage map | 0.12 | 0.04 | 0.17 | 0.04 | 0.10 | 0.03 | 0.09 | 0.03 | 0.19 | 0.02 | 0.09 | 0.03 |
| **BraInCoRL-100** | **0.11** | **0.07** | **0.15** | **0.08** | **0.18** | **0.07** | **0.08** | **0.02** | **0.08** | **0.04** | **0.12** | **0.07** |

Table S.5: **Voxel-wise explained variance with the DINO backbone for Subjects 1 and 2.** We report performance for our in-context model (BraInCoRL), the fully trained oracle ("Fully Trained"), within-subject ridge regression baselines (100, 300), and the FsAverage map across five category-selective regions (faces, places, bodies, words, food).

| | Faces | | Places | | Bodies | | Words | | Food | | Mean | |
|---|---|---|---|---|---|---|---|---|---|---|---|---|
| | **S1** | **S2** | **S1** | **S2** | **S1** | **S2** | **S1** | **S2** | **S1** | **S2** | **S1** | **S2** |
| Fully Trained | 0.15 | 0.13 | 0.16 | 0.22 | 0.24 | 0.20 | 0.08 | 0.08 | 0.13 | 0.14 | 0.14 | 0.15 |
| Ridge-100 | 0.03 | 0.02 | 0.03 | 0.05 | 0.07 | 0.05 | 0.00 | 0.01 | 0.02 | 0.03 | 0.02 | 0.03 |
| Ridge-300 | 0.07 | 0.05 | 0.07 | 0.10 | 0.14 | 0.10 | 0.03 | 0.03 | 0.05 | 0.05 | 0.06 | 0.06 |
| FsAverage map | 0.13 | 0.06 | 0.11 | 0.19 | 0.09 | 0.08 | 0.06 | 0.03 | 0.14 | 0.18 | 0.08 | 0.06 |
| **BraInCoRL-100** | **0.14** | **0.12** | **0.15** | **0.21** | **0.23** | **0.18** | **0.07** | **0.07** | **0.11** | **0.12** | **0.12** | **0.14** |

Table S.6: **Voxel-wise explained variance with the DINO backbone for Subjects 3 and 4.** We report performance for our in-context model (BraInCoRL), the fully trained oracle ("Fully Trained"), within-subject ridge regression baselines (100, 300), and the FsAverage map across five category-selective regions (faces, places, bodies, words, food).

| | Faces | | Places | | Bodies | | Words | | Food | | Mean | |
|---|---|---|---|---|---|---|---|---|---|---|---|---|
| | **S3** | **S4** | **S3** | **S4** | **S3** | **S4** | **S3** | **S4** | **S3** | **S4** | **S3** | **S4** |
| Fully Trained | 0.13 | 0.11 | 0.13 | 0.12 | 0.13 | 0.14 | 0.06 | 0.05 | 0.08 | 0.09 | 0.10 | 0.11 |
| Ridge-100 | 0.02 | 0.00 | 0.03 | 0.01 | 0.03 | 0.02 | -0.01 | -0.02 | 0.00 | -0.02 | 0.01 | 0.01 |
| Ridge-300 | 0.05 | 0.04 | 0.06 | 0.04 | 0.06 | 0.07 | 0.02 | 0.01 | 0.03 | 0.03 | 0.04 | 0.04 |
| FsAverage map | 0.10 | 0.03 | 0.14 | 0.05 | 0.11 | 0.06 | 0.07 | 0.03 | 0.10 | 0.07 | 0.09 | 0.04 |
| **BraInCoRL-100** | **0.11** | **0.10** | **0.13** | **0.12** | **0.13** | **0.14** | **0.05** | **0.05** | **0.06** | **0.08** | **0.09** | **0.10** |

Table S.7: **Voxel-wise explained variance with the DINO backbone for Subjects 5 and 6.** We report performance for our in-context model (BraInCoRL), the fully trained reference ("Fully Trained"), within-subject ridge regression baselines (100, 300), and the FsAverage map across five category-selective regions (faces, places, bodies, words, food).

| | Faces | | Places | | Bodies | | Words | | Food | | Mean | |
|---|---|---|---|---|---|---|---|---|---|---|---|---|
| | **S5** | **S6** | **S5** | **S6** | **S5** | **S6** | **S5** | **S6** | **S5** | **S6** | **S5** | **S6** |
| Fully Trained | 0.19 | 0.14 | 0.27 | 0.11 | 0.22 | 0.15 | 0.13 | 0.06 | 0.20 | 0.06 | 0.19 | 0.08 |
| Ridge-100 | 0.04 | 0.02 | 0.05 | 0.01 | 0.07 | 0.03 | 0.01 | -0.01 | 0.03 | -0.01 | 0.04 | 0.01 |
| Ridge-300 | 0.08 | 0.06 | 0.13 | 0.04 | 0.12 | 0.07 | 0.04 | 0.01 | 0.08 | 0.02 | 0.09 | 0.03 |
| FsAverage map | 0.07 | 0.05 | 0.11 | 0.08 | 0.06 | 0.04 | 0.05 | 0.06 | 0.08 | 0.04 | 0.07 | 0.05 |
| **BraInCoRL-100** | **0.18** | **0.12** | **0.26** | **0.10** | **0.21** | **0.14** | **0.10** | **0.04** | **0.18** | **0.04** | **0.17** | **0.07** |

Table S.8: **Voxel-wise explained variance with the DINO backbone for Subjects 7 and 8.** We report performance for our in-context model (BraInCoRL), the fully trained oracle ("Fully Trained"), within-subject ridge regression baselines (100, 300), and the FsAverage map across five category-selective regions (faces, places, bodies, words, food).

| | Faces | | Places | | Bodies | | Words | | Food | | Mean | |
|---|---|---|---|---|---|---|---|---|---|---|---|---|
| | S7 | S8 | S7 | S8 | S7 | S8 | S7 | S8 | S7 | S8 | S7 | S8 |
| Fully Trained | 0.10 | 0.06 | 0.15 | 0.08 | 0.18 | 0.07 | 0.08 | 0.02 | 0.09 | 0.03 | 0.12 | 0.07 |
| Ridge-100 | 0.02 | -0.00 | 0.02 | 0.00 | 0.06 | 0.00 | 0.01 | -0.01 | 0.00 | -0.00 | 0.02 | -0.00 |
| Ridge-300 | 0.05 | 0.02 | 0.06 | 0.03 | 0.10 | 0.03 | 0.03 | 0.00 | 0.02 | 0.01 | 0.05 | 0.02 |
| FsAverage map | 0.12 | 0.04 | 0.17 | 0.04 | 0.10 | 0.03 | 0.09 | 0.03 | 0.19 | 0.02 | 0.09 | 0.03 |
| **BraInCoRL-100** | **0.10** | **0.06** | **0.14** | **0.07** | **0.17** | **0.07** | **0.07** | **0.02** | **0.07** | **0.04** | **0.10** | **0.06** |

Table S.9: **Voxel-wise explained variance with the SigLIP backbone for Subjects 1 and 2.** We report performance for our in-context model (BraInCoRL), the fully trained oracle ("Fully Trained"), within-subject ridge regression baselines (100, 300), and the FsAverage map across five category-selective regions (faces, places, bodies, words, food).

| | Faces | | Places | | Bodies | | Words | | Food | | Mean | |
|---|---|---|---|---|---|---|---|---|---|---|---|---|
| | S1 | S2 | S1 | S2 | S1 | S2 | S1 | S2 | S1 | S2 | S1 | S2 |
| Fully Trained | 0.19 | 0.17 | 0.21 | 0.27 | 0.30 | 0.25 | 0.12 | 0.11 | 0.17 | 0.18 | 0.19 | 0.20 |
| Ridge-100 | 0.10 | 0.07 | 0.09 | 0.14 | 0.18 | 0.12 | 0.03 | 0.04 | 0.06 | 0.07 | 0.08 | 0.08 |
| Ridge-300 | 0.14 | 0.11 | 0.14 | 0.20 | 0.23 | 0.17 | 0.07 | 0.06 | 0.11 | 0.12 | 0.12 | 0.13 |
| FsAverage map | 0.13 | 0.06 | 0.11 | 0.19 | 0.09 | 0.08 | 0.06 | 0.03 | 0.14 | 0.18 | 0.08 | 0.06 |
| **BraInCoRL-100** | **0.17** | **0.13** | **0.18** | **0.24** | **0.27** | **0.21** | **0.09** | **0.08** | **0.13** | **0.14** | **0.15** | **0.16** |

Table S.10: **Voxel-wise explained variance with the SigLIP backbone for Subjects 3 and 4.** We report performance for our in-context model (BraInCoRL), the fully trained oracle ("Fully Trained"), within-subject ridge regression baselines (100, 300), and the FsAverage map across five category-selective regions (faces, places, bodies, words, food).

| | Faces | | Places | | Bodies | | Words | | Food | | Mean | |
|---|---|---|---|---|---|---|---|---|---|---|---|---|
| | S3 | S4 | S3 | S4 | S3 | S4 | S3 | S4 | S3 | S4 | S3 | S4 |
| Fully Trained | 0.17 | 0.14 | 0.17 | 0.17 | 0.18 | 0.18 | 0.10 | 0.08 | 0.11 | 0.13 | 0.14 | 0.15 |
| Ridge-100 | 0.07 | 0.04 | 0.08 | 0.06 | 0.09 | 0.07 | 0.02 | 0.01 | 0.03 | 0.04 | 0.05 | 0.05 |
| Ridge-300 | 0.11 | 0.08 | 0.11 | 0.10 | 0.12 | 0.12 | 0.05 | 0.03 | 0.06 | 0.07 | 0.08 | 0.09 |
| FsAverage map | 0.10 | 0.03 | 0.14 | 0.05 | 0.11 | 0.06 | 0.07 | 0.03 | 0.10 | 0.07 | 0.10 | 0.04 |
| **BraInCoRL-100** | **0.12** | **0.11** | **0.13** | **0.14** | **0.14** | **0.15** | **0.05** | **0.05** | **0.06** | **0.09** | **0.10** | **0.12** |

Table S.11: **Voxel-wise explained variance with the SigLIP backbone for Subjects 5 and 6.** We report performance for our in-context model (BraInCoRL), the fully trained reference ("Fully Trained"), within-subject ridge regression baselines (100, 300), and the FsAverage map across five category-selective regions (faces, places, bodies, words, food).

| | Faces | | Places | | Bodies | | Words | | Food | | Mean | |
|---|---|---|---|---|---|---|---|---|---|---|---|---|
| | S5 | S6 | S5 | S6 | S5 | S6 | S5 | S6 | S5 | S6 | S5 | S6 |
| Fully Trained | 0.24 | 0.18 | 0.33 | 0.15 | 0.28 | 0.19 | 0.18 | 0.10 | 0.26 | 0.10 | 0.24 | 0.12 |
| Ridge-100 | 0.11 | 0.07 | 0.16 | 0.04 | 0.14 | 0.09 | 0.06 | 0.01 | 0.11 | 0.02 | 0.11 | 0.04 |
| Ridge-300 | 0.17 | 0.12 | 0.24 | 0.08 | 0.20 | 0.14 | 0.10 | 0.04 | 0.17 | 0.05 | 0.16 | 0.07 |
| FsAverage map | 0.07 | 0.05 | 0.11 | 0.08 | 0.06 | 0.04 | 0.05 | 0.06 | 0.08 | 0.04 | 0.07 | 0.05 |
| **BraInCoRL-100** | **0.20** | **0.14** | **0.28** | **0.11** | **0.23** | **0.16** | **0.12** | **0.05** | **0.19** | **0.05** | **0.19** | **0.09** |

Table S.12: **Voxel-wise explained variance with the SigLIP backbone for Subjects 7 and 8.**
We report performance for our in-context model (BraInCoRL), the fully trained oracle ("Fully
Trained"), within-subject ridge regression baselines (100, 300), and the FsAverage map across five
category-selective regions (faces, places, bodies, words, food).

| | Faces | | Places | | Bodies | | Words | | Food | | Mean | |
|---|---|---|---|---|---|---|---|---|---|---|---|---|
| | **S7** | **S8** | **S7** | **S8** | **S7** | **S8** | **S7** | **S8** | **S7** | **S8** | **S7** | **S8** |
| Fully Trained | 0.14 | 0.09 | 0.15 | 0.10 | 0.22 | 0.09 | 0.13 | 0.04 | 0.13 | 0.06 | 0.16 | 0.09 |
| Ridge-100 | 0.06 | 0.02 | 0.08 | 0.04 | 0.11 | 0.03 | 0.03 | 0.00 | 0.03 | 0.02 | 0.06 | 0.03 |
| Ridge-300 | 0.10 | 0.05 | 0.12 | 0.07 | 0.15 | 0.05 | 0.06 | 0.02 | 0.06 | 0.03 | 0.09 | 0.05 |
| FsAverage map | 0.12 | 0.04 | 0.17 | 0.04 | 0.10 | 0.03 | 0.09 | 0.03 | 0.19 | 0.02 | 0.09 | 0.03 |
| **BraInCoRL-100** | **0.11** | **0.07** | **0.15** | **0.08** | **0.19** | **0.08** | **0.08** | **0.02** | **0.09** | **0.04** | **0.12** | **0.07** |

### A.7 Voxelwise explained variance across varying support set sizes for more subjects, backbones and for pretrain-only models

In this section, we investigate how the size of the in-context support set affects voxelwise predictive performance. We evaluate three image-encoding backbones (CLIP, DINO, SigLIP) on eight subjects (S1–S8) by comparing the performance of BraInCoRL with the pretrain-only BraInCoRL (i.e. same architecture but only pretrained), the within-subject ridge-regression baseline, and the fully trained reference model fit to converge on each subject's full 9,000-image training set.

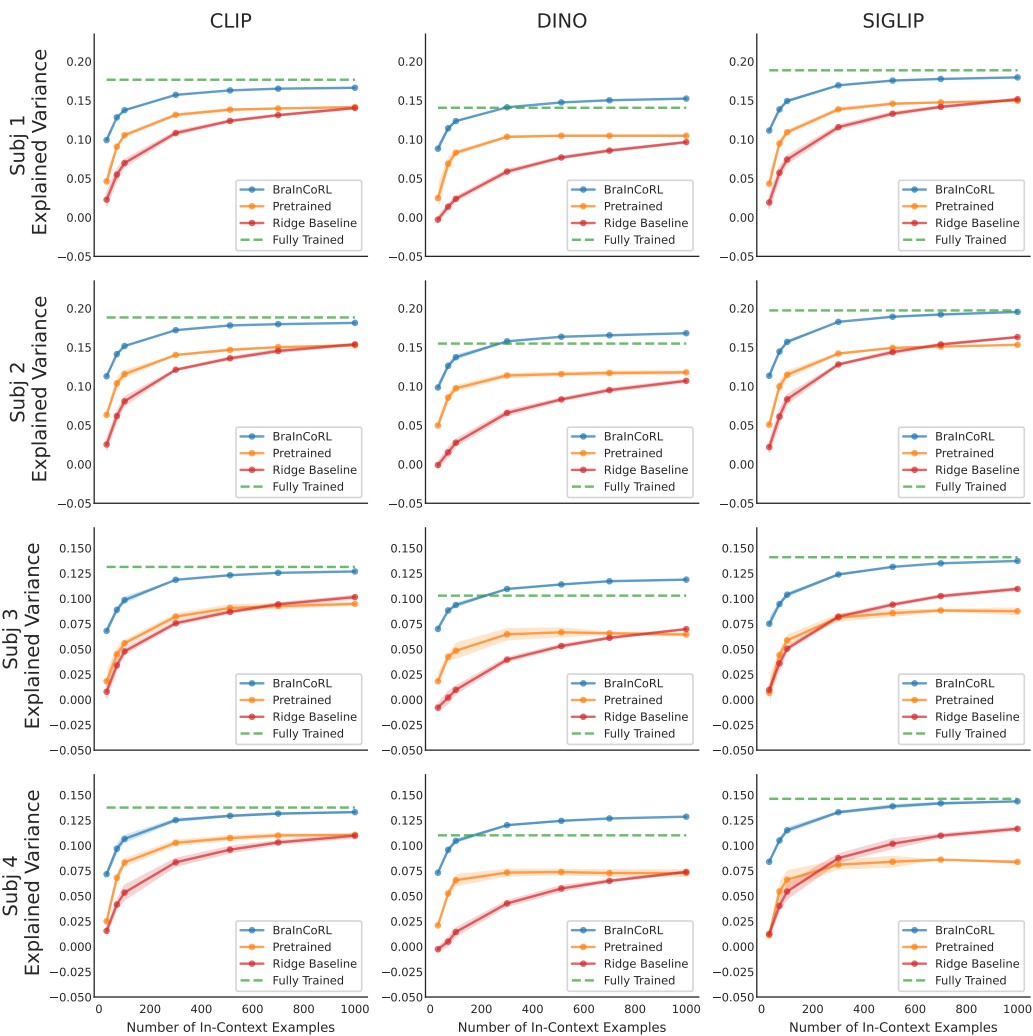

Figure S.7: **Voxelwise explained variance as a function of in-context support set size.** Voxelwise explained variance is visualized for subjects 1, 2, 3, 4 for each backbone (CLIP, DINO, SigLIP). we plot results for the BraInCoRL, along with the pretrain-only BraInCoRL (i.e. same architecture but only pretrained), the within-subject ridge-regression baseline, and the fully trained reference model using all 9,000 images.

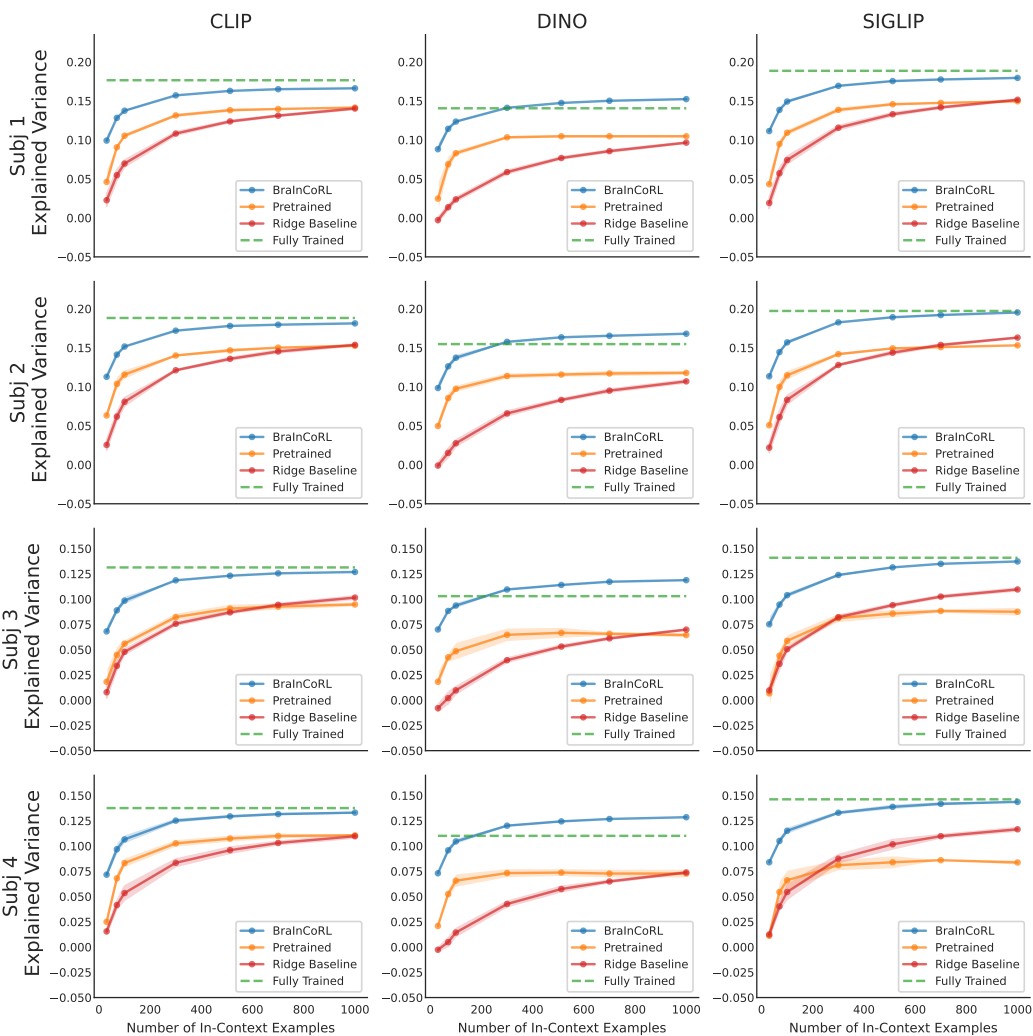

Figure S.8: **Voxelwise explained variance as a function of in-context support set size.** Voxelwise explained variance is visualized for subjects 5, 6, 7, 8 for each backbone (CLIP, DINO, SigLIP). we plot results for the BraInCoRL, along with the pretrain-only BraInCoRL (i.e. same architecture but only pretrained), the within-subject ridge-regression baseline, and the fully trained reference model using all 9,000 images.

### A.8 Impact of holding out the test subject's unique images during meta-training evaluated on more backbones

In this section, we further conduct ablations by evaluating a BraInCoRL model trained without holding out the test subject's support images ("BraInCoRL no HO") and a BraInCoRL model with only pretraining, on DINO and SigLIP backbones. The result indicates that fine-tuning on real neural data enhances performance, and that BraInCoRL is able to generalize effectively to entirely unseen images without having encountered them during training.

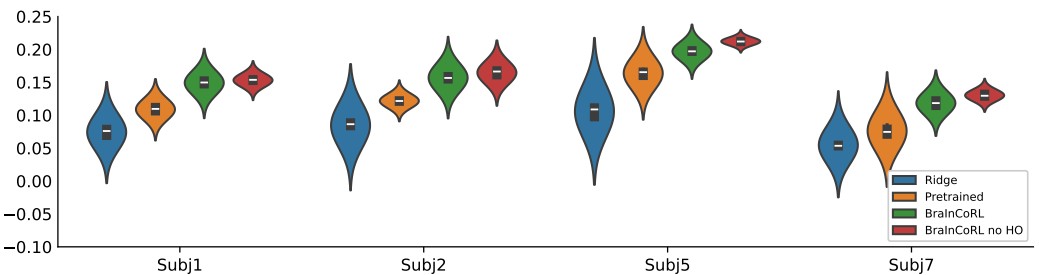

Figure S.9: **Distributions of voxelwise explained variance for subjects 1, 2, 5 and 7 using DINO encoding.** Results confirm that finetuning with real neural data boosts performance and that BraInCoRL can generalize well to previously unseen images without requiring them during training.

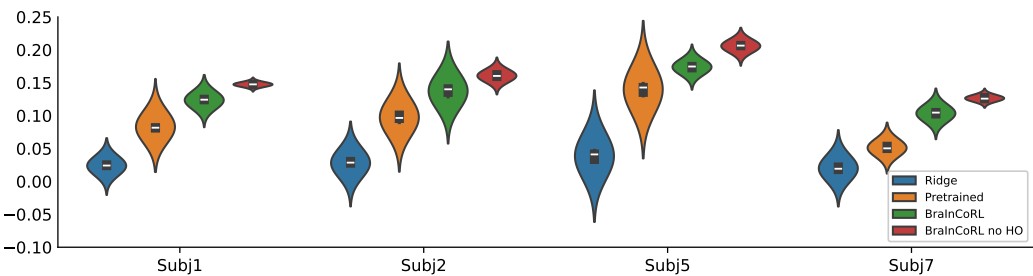

Figure S.10: **Distributions of voxelwise explained variance for subjects 1, 2, 5 and 7 using SigLIP encoding.** Results confirm that finetuning with real neural data boosts performance and that BraInCoRL can generalize well to previously unseen images without requiring them during training.

## A.9 Correlation of each backbone's predictions with fully trained activation predictions

Using various image-encoding backbones, we plot how each subject's BraInCoRL predicted explained variance correlates with the fully trained model's explained variance (fully trained model refers to the fully trained reference model fit to converge on each subject's full 9,000-image training set). Across every backbone and all subjects, this correlation remains uniformly high.

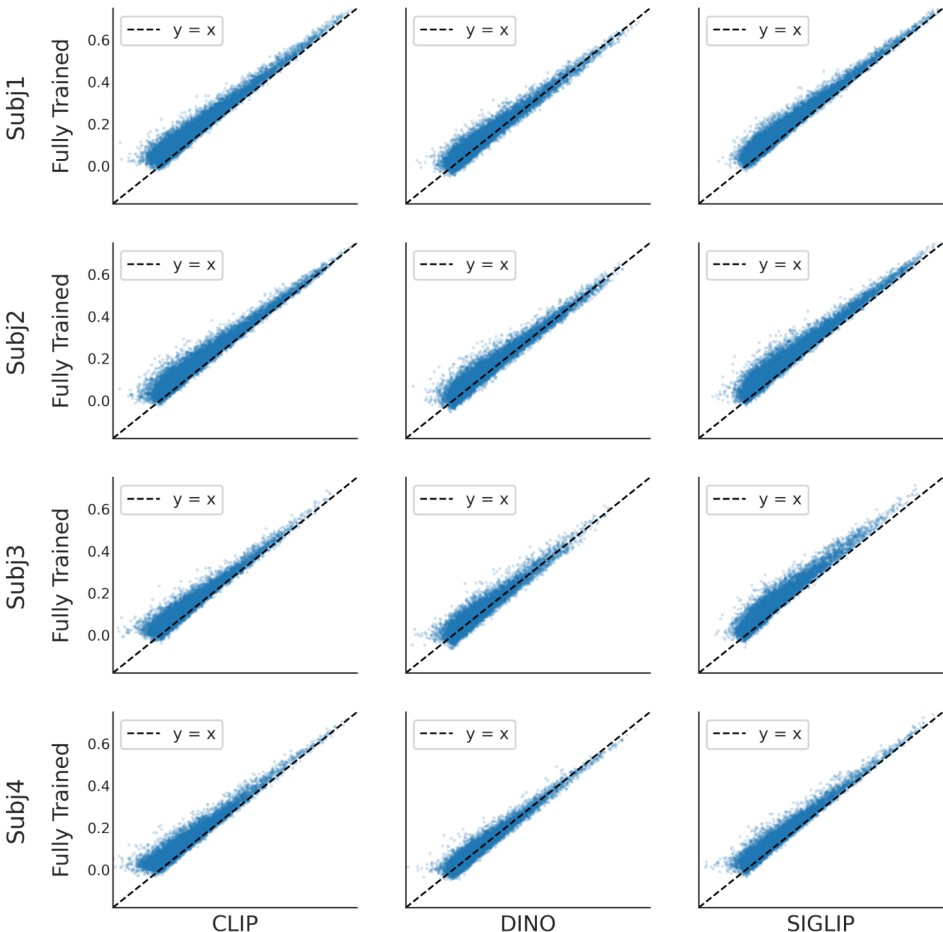

Figure S.11: **Voxelwise explained-variance correlation across backbones for subject 1, 2, 3, 4.** Each panel shows a scatter of the BraInCoRL's explained-variance predictions (100 in-context examples) versus the fully trained reference model's explained variance for each voxel. Rows correspond to subjects (1, 2, 3, 4) and columns to image-encoding backbones (CLIP, DINO, SigLIP). The dashed line marks $y = x$.

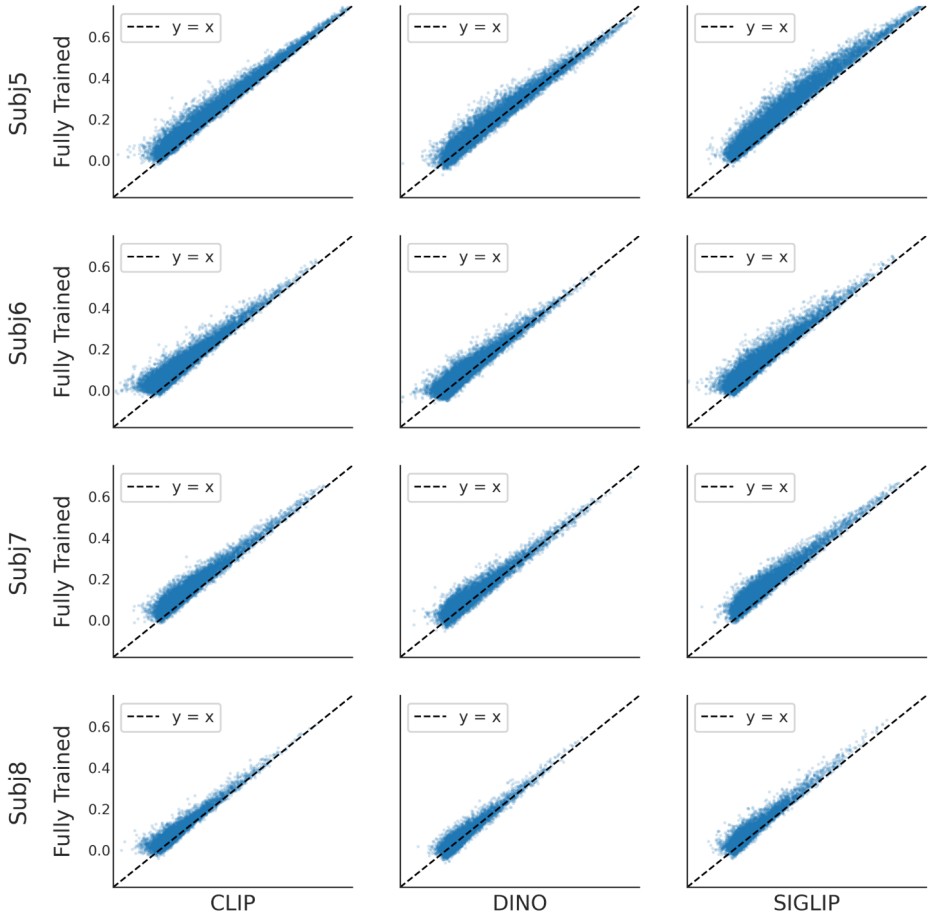

Figure S.12: **Voxelwise explained-variance correlation across backbones for subject 5, 6, 7, 8.** Each panel shows a scatter of the BraInCoRL's explained-variance predictions (100 in-context examples) versus the fully trained reference model's explained variance for each voxel. Rows correspond to subjects (5, 6, 7, 8) and columns to image-encoding backbones (CLIP, DINO, SigLIP). The dashed line marks $y = x$.

## A.10 Voxelwise explained-variance evaluation in BOLD5000 for more subjects and different backbones

In this section, we analyze how varying the number of in-context examples influences voxel-level prediction performance on the BOLD5000 dataset. For subject S2 and S3, we plot the mean Pearson's $r$ between model-predicted and actual BOLD responses as a function of support-set size, comparing our BraInCoRL model against a ridge regression baseline. Results are averaged over five cross-validation folds. Across all three image-encoding backbones (CLIP, DINO, and SIGLIP), BraInCoRL consistently outperforms ridge regression.

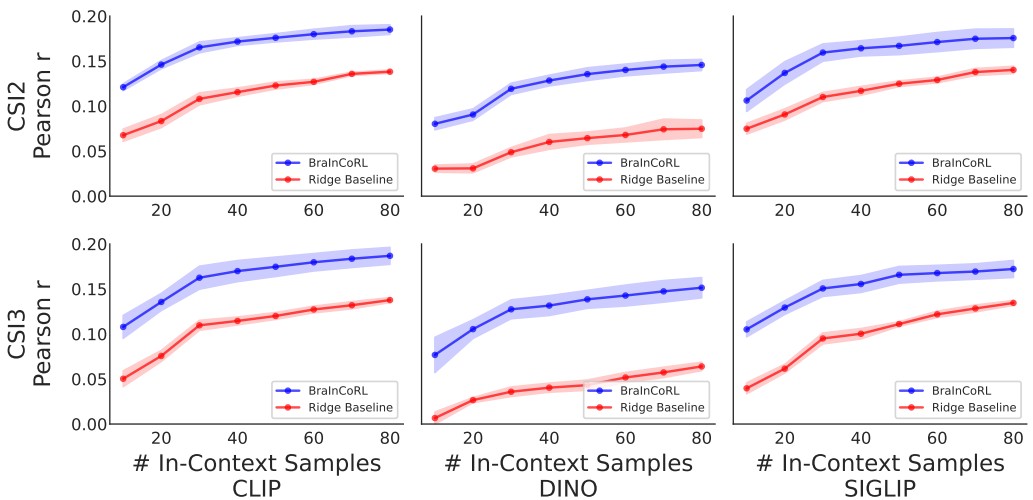

Figure S.13: **Support-set size versus voxelwise Pearson $r$ in BOLD5000.** Each panel shows the mean voxelwise Pearson correlation between predicted and actual BOLD5000 responses for BraInCoRL and ridge regression, plotted against the number of in-context samples.

## A.11 Dimensional reduction of predicted response function weights on more subjects

In this section, we utilize UMAP to visualize the BraInCoRL-predicted voxelwise weights under the CLIP image encoding backbone for subject S1-S8. The cortical maps show color-coded mappings that align well with functionally-defined regions.

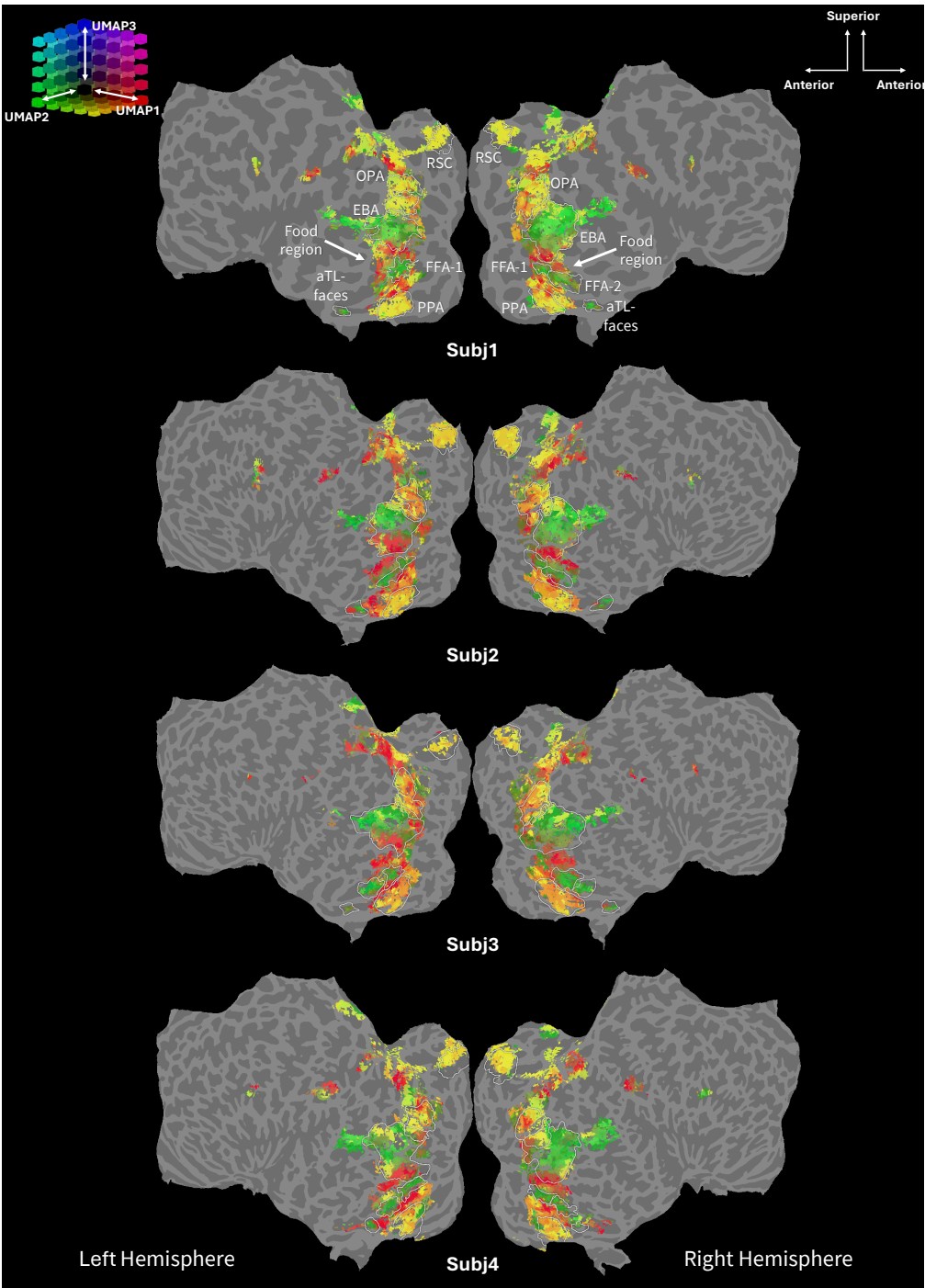

Figure S.14: **Dimensional reduction of predicted response weights for subject S1-S4 under CLIP backbone.** The cortical maps show color-coded mappings that align well with functionally-defined regions: body and face regions (EBA and FFA/aTL-faces), place regions (RSC/OPA/PPA), and food regions (in red).

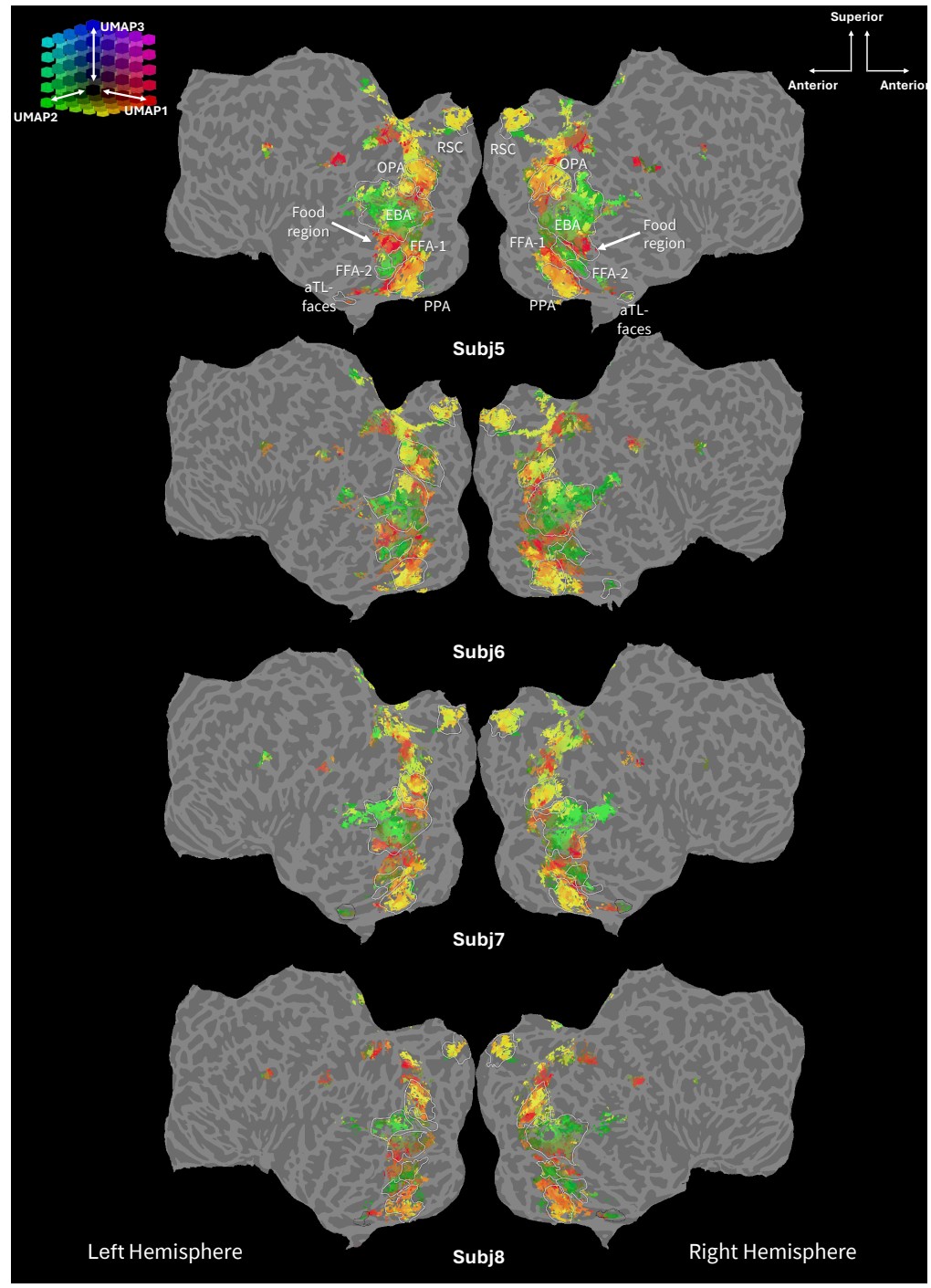

Figure S.15: **Dimensional reduction of predicted response weights for subject S5-S8 under CLIP backbone.** The cortical maps show color-coded mappings that align well with functionally-defined regions: body and face regions (EBA and FFA/aTL-faces), place regions (RSC/OPA/PPA), and food regions (in red).

## A.12 Predicting cortical responses from natural language prompts on more subjects

In this section, we further predict cortical responses from natural language prompts on subject 2-8. For each semantic category, we convert a natural language prompt into a CLIP text embedding, project it into the image feature space, and use BraInCoRL to predict the corresponding voxel activation map. The predicted activations align closely with known $t$-statistic of category-selective region, illustrating the potential for zero-shot, language-driven functional mapping of visual cortex.

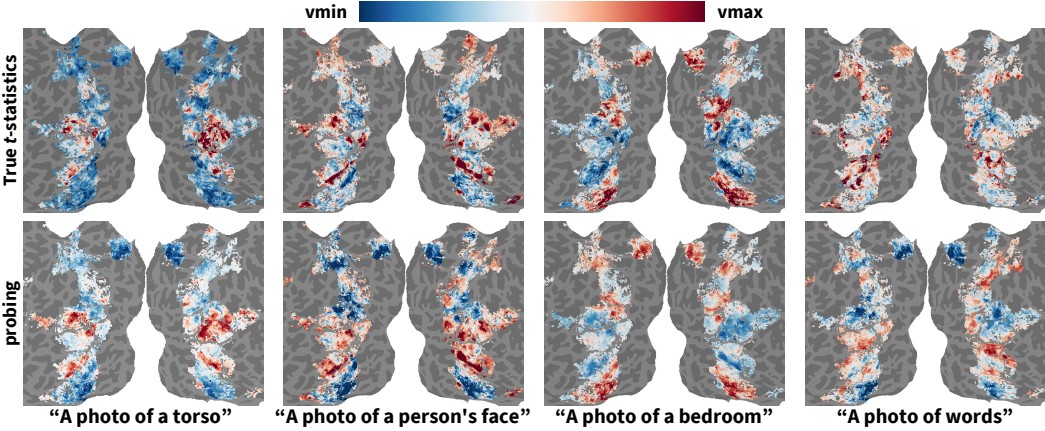

Figure S.16: **Predicting responses of natural language prompts for subject 2.** We convert each text prompt corresponding to a semantic category into a CLIP text embedding, project it into image-feature space, and predict its cortical activation on subject 2. The resulting activation maps closely match the $t$-statistics of known category-selective regions, demonstrating the feasibility of language-driven, zero-shot functional mapping of the visual cortex.

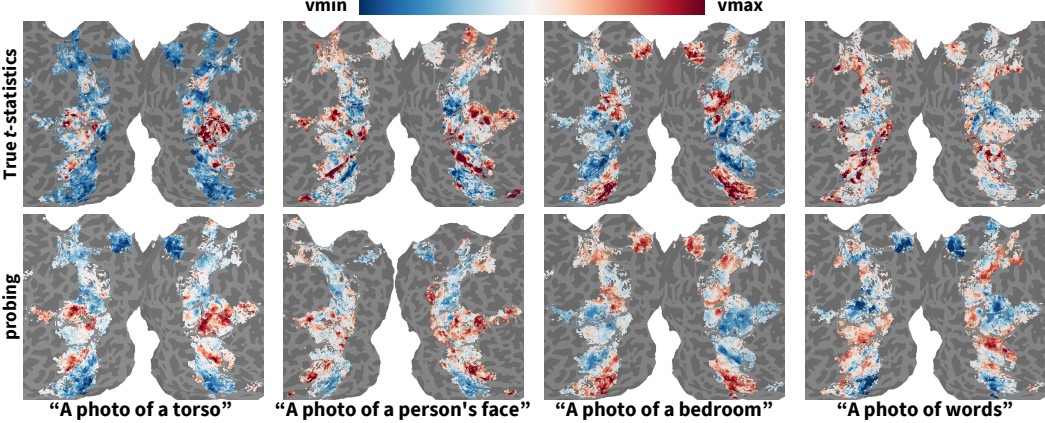

Figure S.17: **Predicting responses of natural language prompts for subject 3.** We convert each text prompt corresponding to a semantic category into a CLIP text embedding, project it into image-feature space, and predict its cortical activation on subject 3. The resulting activation maps closely match the $t$-statistics of known category-selective regions, demonstrating the feasibility of language-driven, zero-shot functional mapping of the visual cortex.

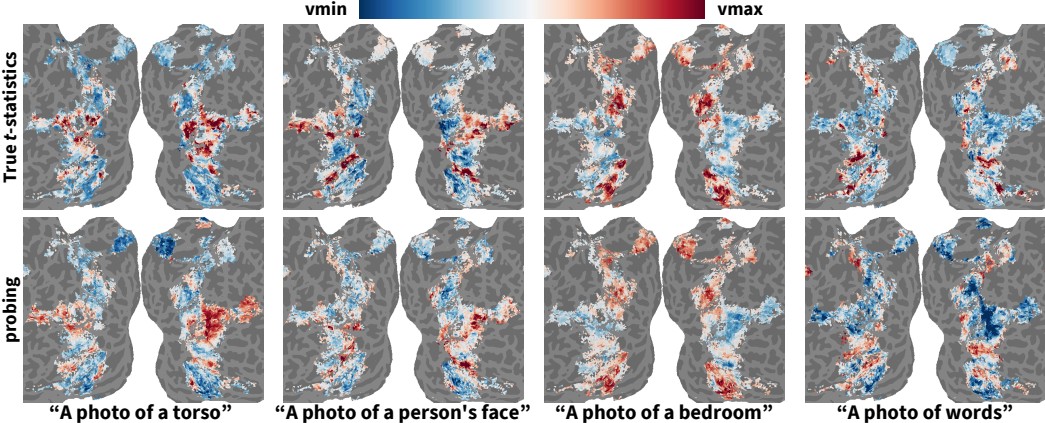

Figure S.18: **Predicting responses of natural language prompts for subject 4.** We convert each text prompt corresponding to a semantic category into a CLIP text embedding, project it into image-feature space, and predict its cortical activation on subject 4. The resulting activation maps closely match the *t*-statistics of known category-selective regions, demonstrating the feasibility of language-driven, zero-shot functional mapping of the visual cortex.

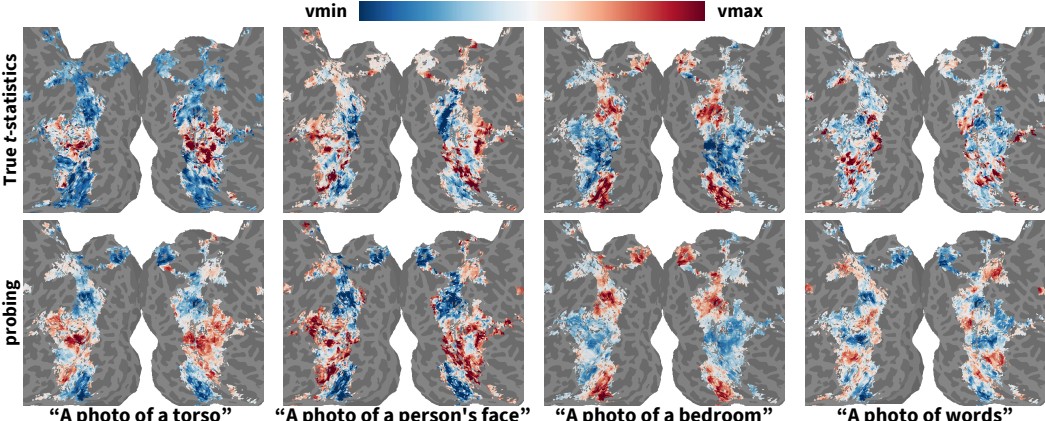

Figure S.19: **Predicting responses of natural language prompts for subject 5.** We convert each text prompt corresponding to a semantic category into a CLIP text embedding, project it into image-feature space, and predict its cortical activation on subject 5. The resulting activation maps closely match the *t*-statistics of known category-selective regions, demonstrating the feasibility of language-driven, zero-shot functional mapping of the visual cortex.

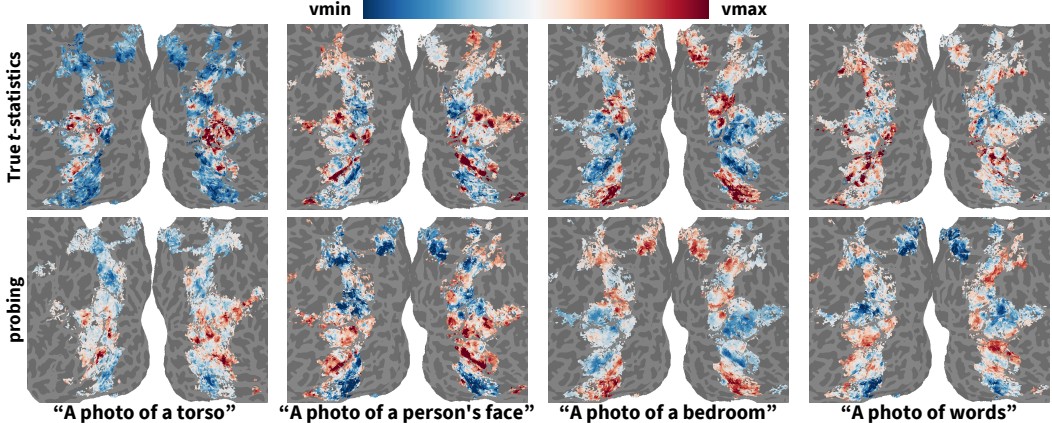

Figure S.20: **Predicting responses of natural language prompts for subject 6.** We convert each text prompt corresponding to a semantic category into a CLIP text embedding, project it into image-feature space, and predict its cortical activation on subject 6. The resulting activation maps closely match the $t$-statistics of known category-selective regions, demonstrating the feasibility of language-driven, zero-shot functional mapping of the visual cortex.

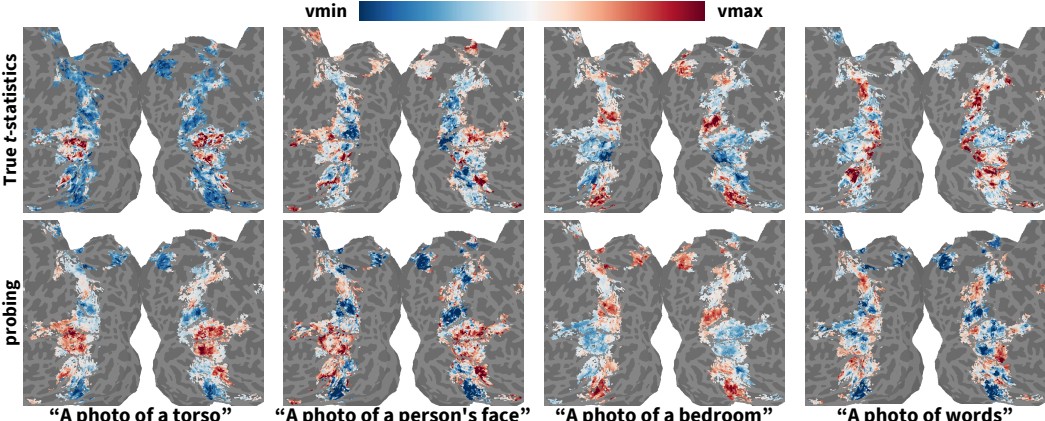

Figure S.21: **Predicting responses of natural language prompts for subject 7.** We convert each text prompt corresponding to a semantic category into a CLIP text embedding, project it into image-feature space, and predict its cortical activation on subject 7. The resulting activation maps closely match the $t$-statistics of known category-selective regions, demonstrating the feasibility of language-driven, zero-shot functional mapping of the visual cortex.

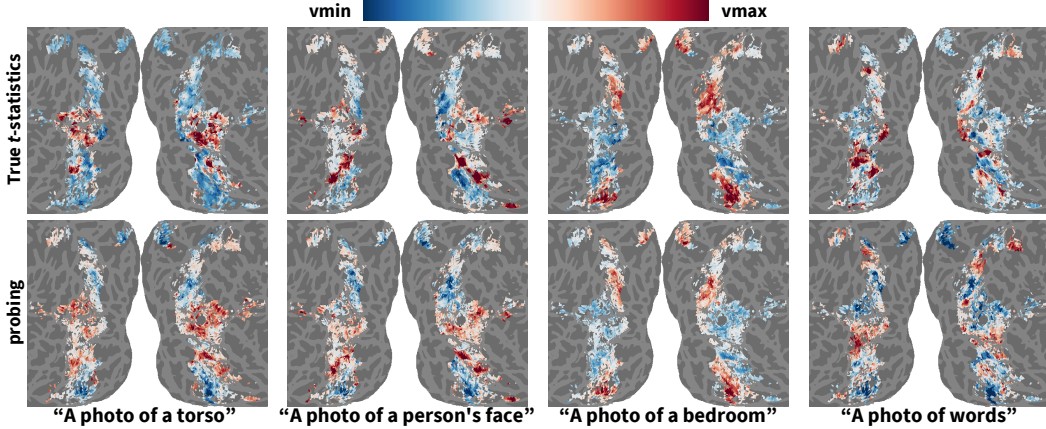

Figure S.22: **Predicting responses of natural language prompts for subject 8.** We convert each text prompt corresponding to a semantic category into a CLIP text embedding, project it into image-feature space, and predict its cortical activation on subject 8. The resulting activation maps closely match the *t*-statistics of known category-selective regions, demonstrating the feasibility of language-driven, zero-shot functional mapping of the visual cortex.

### A.13 Voxelwise prompt classification accuracy for more subjects

In this section, we further quantify the semantic specificity of BraInCoRL's voxelwise predictions on subject 3-8. we compute, for each subject and each category-selective ROI, the fraction of voxels whose peak predicted activation corresponded to the semantic category named by the text prompt.

Table S.13: **Voxelwise prompt classification accuracy for subjects 3 and 4.** Each cell shows the percentage of voxels in a given category selective region (columns) whose peak predicted activation was elicited by a specific semantic prompt (rows, see Appendix) for subject 3 and 4. Using only 100 support images, BraInCoRL effectively localizes category-selective regions with high data efficiency.

|        | Bodies |      | Faces |      | Places |      | Food |      | Words |      |
|--------|--------|------|-------|------|--------|------|------|------|-------|------|
|        | **S3** | **S4** | **S3** | **S4** | **S3** | **S4** | **S3** | **S4** | **S3** | **S4** |
| Bodies | **0.57** | **0.42** | 0.23 | 0.12 | 0.04 | 0.03 | 0.16 | 0.20 | 0.12 | 0.19 |
| Faces  | 0.29 | 0.36 | **0.60** | **0.66** | 0.02 | 0.03 | 0.11 | 0.05 | 0.20 | 0.12 |
| Places | 0.04 | 0.08 | 0.02 | 0.06 | **0.84** | **0.82** | 0.15 | 0.20 | 0.08 | 0.16 |
| Food   | 0.07 | 0.09 | 0.14 | 0.12 | 0.09 | 0.09 | **0.53** | **0.51** | **0.51** | **0.43** |
| Words  | 0.02 | 0.04 | 0.01 | 0.04 | 0.01 | 0.02 | 0.05 | 0.05 | 0.09 | 0.10 |

Table S.14: **Voxelwise prompt classification accuracy for subjects 5 and 6.** Each cell shows the percentage of voxels in a given category selective region (columns) whose peak predicted activation was elicited by a specific semantic prompt (rows, see Appendix) for subject 5 and 6. Using only 100 support images, BraInCoRL effectively localizes category-selective regions with high data efficiency.

|        | Bodies |      | Faces |      | Places |      | Food |      | Words |      |
|--------|--------|------|-------|------|--------|------|------|------|-------|------|
|        | **S5** | **S6** | **S5** | **S6** | **S5** | **S6** | **S5** | **S6** | **S5** | **S6** |
| Bodies | **0.54** | **0.64** | 0.17 | 0.21 | 0.01 | 0.08 | 0.19 | 0.15 | 0.27 | 0.25 |
| Faces  | 0.29 | 0.25 | **0.65** | **0.63** | 0.00 | 0.04 | 0.03 | 0.05 | 0.20 | 0.15 |
| Places | 0.06 | 0.01 | 0.05 | 0.01 | **0.88** | **0.65** | 0.13 | 0.09 | 0.10 | 0.04 |
| Food   | 0.09 | 0.07 | 0.13 | 0.11 | 0.10 | 0.20 | **0.64** | **0.66** | **0.39** | **0.41** |
| Words  | 0.02 | 0.03 | 0.01 | 0.04 | 0.00 | 0.04 | 0.01 | 0.06 | 0.05 | 0.15 |

Table S.15: **Voxelwise prompt classification accuracy for subjects 7 and 8.** Each cell shows the percentage of voxels in a given category selective region (columns) whose peak predicted activation was elicited by a specific semantic prompt (rows, see Appendix) for subject 7 and 8. Using only 100 support images, BraInCoRL effectively localizes category-selective regions with high data efficiency.

|        | Bodies |      | Faces |      | Places |      | Food |      | Words |      |
|--------|--------|------|-------|------|--------|------|------|------|-------|------|
|        | **S7** | **S8** | **S7** | **S8** | **S7** | **S8** | **S7** | **S8** | **S7** | **S8** |
| Bodies | **0.69** | **0.57** | 0.26 | 0.15 | 0.01 | 0.07 | 0.08 | 0.20 | 0.19 | 0.17 |
| Faces  | 0.19 | 0.25 | **0.59** | **0.59** | 0.01 | 0.04 | 0.04 | 0.10 | 0.22 | 0.18 |
| Places | 0.04 | 0.05 | 0.02 | 0.03 | **0.89** | **0.58** | 0.12 | 0.06 | 0.13 | 0.05 |
| Food   | 0.06 | 0.10 | 0.11 | 0.20 | 0.07 | 0.26 | **0.68** | **0.57** | **0.32** | **0.47** |
| Words  | 0.02 | 0.03 | 0.03 | 0.03 | 0.01 | 0.04 | 0.07 | 0.08 | 0.14 | 0.12 |

### A.14    Additional evaluation of BraInCoRL on NSD dataset

In this section, we provide two more evaluation metrics, namely Pearson $R$, and Spearman's rank correlation coefficient (Spearman's $\rho$) for NSD dataset on Subject 1. In this case, the BraInCoRL model has not been trained or finetuned on Subject 1, while the Fully Trained model is trained on 9,000 images from this subject.

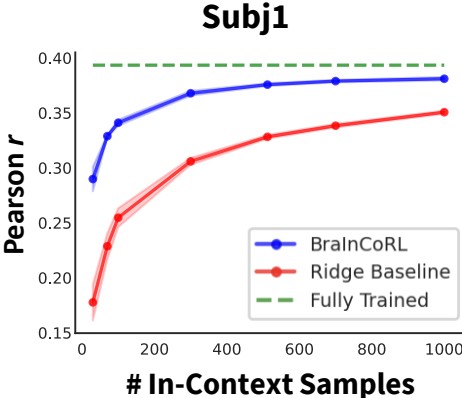

Figure S.23: Voxel-wise Pearson $R$ for BraInCoRL, within-subject ridge regression baseline and fully-trained reference model (NSD dataset, CLIP backbone, Subject 1, higher is better).

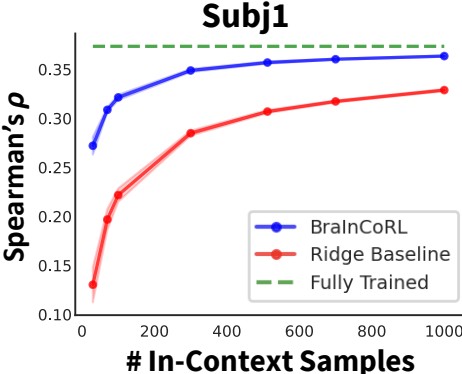

Figure S.24: Voxel-wise Spearman's $\rho$ for BraInCoRL, within-subject ridge regression baseline and fully-trained reference model (NSD dataset, CLIP backbone, Subject 1, higher is better).

### A.15 Additional evaluation of BraInCoRL on BOLD5000 dataset

In this section, we provide two more evaluation metrics, namely explained variance and Spearman's rank correlation coefficient (Spearman's $\rho$) for BOLD5000 dataset on Subject CSI1.

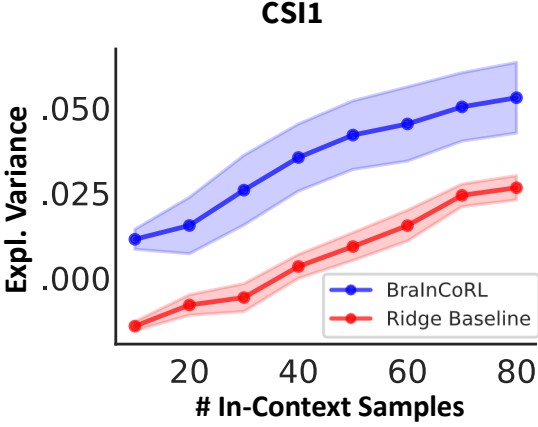

Figure S.25: Voxel-wise explained variance for BraInCoRLand within-subject ridge regression baseline (BOLD5000 dataset, CLIP backbone, Subject CSI1, higher is better).

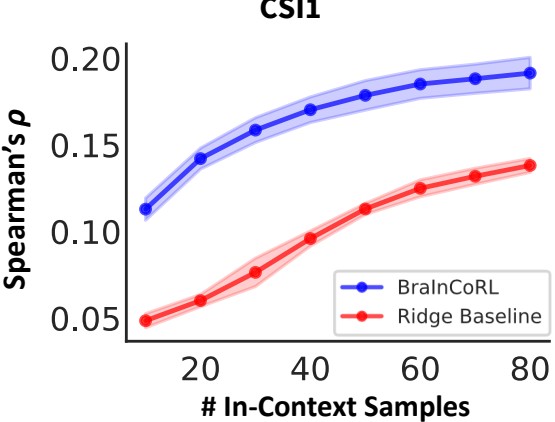

Figure S.26: Voxel-wise Spearman's $\rho$ for BraInCoRLand within-subject ridge regression baseline (BOLD5000 dataset, CLIP backbone, Subject CSI1, higher is better).

## A.16 Evaluation of each training stage's contribution

In this section, we present the voxelwise explained variance for BraInCoRLtrained until different training stages. The results show that the progression from synthetic foundation → context flexibility → biological adaptation ensures that each fundamental challenge, namely response function coverage, variable context handling, and biological realism, is systematically addressed in the optimal order.

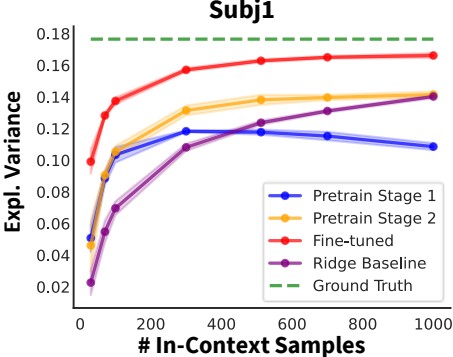

Figure S.27: Voxelwise explained variance for BraInCoRLwith CLIP backbone for NSD Subject 1 on different training stages, compared with ridge baseline and fully trained reference model (higher is better).

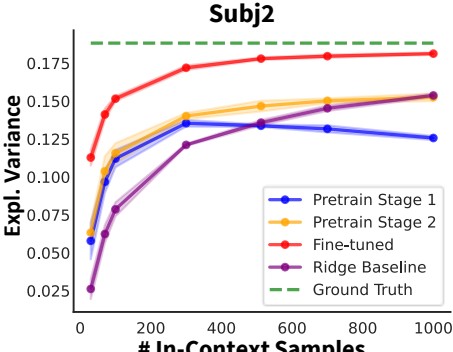

Figure S.28: Voxelwise explained variance for BraInCoRLwith CLIP backbone for NSD Subject 2 on different training stages, compared with ridge baseline and fully trained reference model (higher is better).

### A.17 Performance of BraInCoRL conditioned on the full 9000-image set

In this section, we evaluate the performance of BraInCoRL with 9,000 in-context samples, for the four NSD subjects (S1, S2, S5, S7) focues by our main paper. The performance difference is less than 1% across all subjects when compared to a fully trained model (which is fit to converge on each subject's entire 9,000-image training set using gradient descent over multiple epochs). This means BraInCoRL achieves 94-99% of the fully trained model's performance.

Table S.16: Voxel-wise explained variance of BraInCoRL with the CLIP backbone compared with the fully trained reference model. The difference variance explained is less than 1%.

| Method | Subject | | | |
| --- | --- | --- | --- | --- |
| | S1 | S2 | S5 | S7 |
| Fully Trained | 0.1765 | 0.1882 | 0.2310 | 0.1554 |
| BrainCoRL | **0.1667** | **0.1817** | **0.2225** | **0.1541** |

## A.18 Evaluation on the choice of loss function during training

In this section, we conducted an ablation study on the choice of different loss functions during the finetuning stage optimization and evaluated the model performance.

Our experimental results show that MSE and L1 losses achieve similar performance across all context sizes, with minimal differences. This suggests that both metrics are equally effective for capturing voxelwise neural response patterns.

In addition, the hybrid loss of $0.5 \times \text{MSE loss} + 0.5 \times (1 - \text{cosine similarity})$ underperforms by approximately 2-4% compared to MSE/L1. We argue this is because although cosine similarity captures directional relationships between predicted and true responses, this additional constraint limit the model's ability to accurately predict response magnitudes.

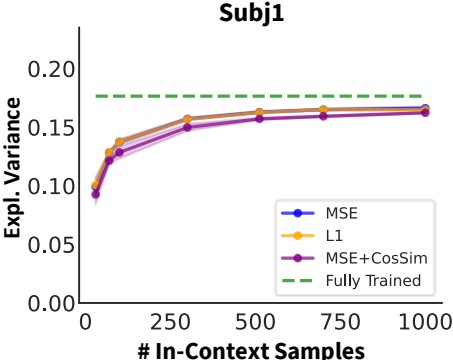

Figure S.29: Voxel-wise explained variance of different training losses with the CLIP backbone for Subject 1 (higher is better).

### A.19 Ablation on logit scaling

In this section, we provide an additional ablation study to evaluate the effect of the logit-scaling, where we report the Voxel-wise explained variance of our BrainCoRL model and the model with the exact same structure but without logit scaling. It is shown that the logit scaling significantly boosts the model performance and generizability of various in-context support set sizes.

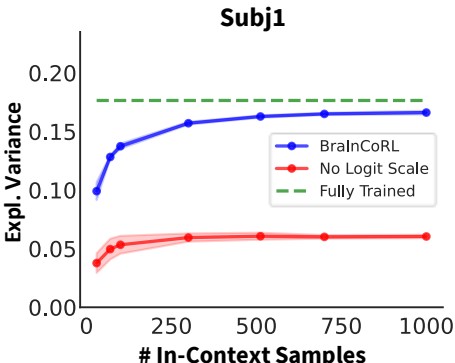

Figure S.30: Voxel-wise explained variance for BrainCoRL with CLIP backbone for Subj 1, compared to the same model architecture but without logit scaling (higher is better).

# Appendix References

[1] Jianlin Su. Analyzing the scale operation of attention from the perspective of entropy invariance, Dec 2021. URL https://kexue.fm/archives/8823.

[2] David Chiang and Peter Cholak. Overcoming a theoretical limitation of self-attention. *arXiv preprint arXiv:2202.12172*, 2022.

[3] Jinze Bai, Shuai Bai, Yunfei Chu, Zeyu Cui, Kai Dang, Xiaodong Deng, Yang Fan, Wenbin Ge, Yu Han, Fei Huang, et al. Qwen technical report. *arXiv preprint arXiv:2309.16609*, 2023.

[4] Meta AI. The llama 4 herd: The beginning of a new era of natively multimodal ai innovation. 4(7):2025, 2025. URL https://ai.meta.com/blog/llama-4-multimodal-intelligence/,checkedon.

