# OpenReview forum: "Meta-Learning an In-Context Transformer Model of Human Higher Visual Cortex"
_NeurIPS.cc/2025/Conference — NeurIPS 2025 poster_

### Official Review · Reviewer_CYvz · 2025-06-29

**Clarity:** 4
**Significance:** 3
**Originality:** 3
**Rating:** 5
**Confidence:** 3

**Summary:**

The authors propose BraInCoRL as a model for predicting neuronal fMRI responses. The approach uses a transformer-based model with meta-learning across voxels and subjects, and few-shot in-context learning across stimuli without the need for finetuning. The authors show that the model generalizes across subjects, stimuli and recording acquisition parameters, outperforming anatomical and subject-wise baselines. They also explore attention patterns and carry out ablations as well as link visual cortex activity to natural language.

**Questions:**

1. The authors assume that the unobserved true response function is noiseless, i.e. all the noise is part of the measurement process. In reality there will be different noise components, some of which will be inherent to the brain response. Could it be useful to model these different sources in more detail?

2. The visual response function is designed and required to be differentiable. How much of a limitation is this?

3. Line 138: The authors used MSE optimization. Did they consider other error metrics?

**Ethical Concerns:**

["NO or VERY MINOR ethics concerns only"]

**Final Justification:**

I had minor concerns to begin with. The authors answered all of my questions and conducted additional analyses, confirming my initial assessment. For these reasons, I keep my score and recommend acceptance.

**Limitations:**

Limitations of the study are discussed in a dedicated section and mention static vs. dynamic stimuli, as well as dataset diversity. Societal impacts are discussed in an appendix section. In my opinion, these sections address all reasonable points.

**Paper Formatting Concerns:**

None.

**Quality:**

4

**Strengths And Weaknesses:**

Overall, this study presents a solid modelling framework for fMRI responses. The methods are well evaluated and different applications are demonstrated. Minor weaknesses include lack of justifications of design decisions. My questions further elaborate on this.

Quality:
The general problem is formalized cleanly. The authors perform a rigorous evaluation, making proper use of held-out data, and performing ablations. Baseline models are well selected and demonstrate the benefits of the proposed model and training, where no finetuning is required.

Clarity:
The paper is very well written. Figures and illustrations are clean and informative. The appendix also provides extensive methodological details on the proposed model and training. Minor points:
Eq. 4: redefines $c$ to represent context length. It might be better to use another symbol here to avoid confusion. Figure 7 should label the y-axis.

Significance:
The authors formulate the encoding problem as a meta-learning task, which addresses the problem that previous encoders were subject specific, requiring large individual datasets. The proposed model and training techniques demonstrate the generalization capabilities of modern models and training. Similar techniques might be applicable to other recording modalities or domains. The main immediate beneficiaries are practitioners using fMRI encoding models.

Originality:
There are previous studies on multi-subject approaches, as cited by the paper [30, 32], which do require finetuning. So the main advance of this work is the complete absence of needing finetuning. The authors adopt many techniques that were developed for LLMs, such as analysis-by-synthesis and making use of logit scaling for variable context length. These are adapted without proposing entirely new techniques but are novel in the context of fMRI encoders. Contrary to previous approaches, each voxel has a response function that is addressed in a meta-learning task. A transformer generates parameters of the visual response functions based on context tokens.

---

> ### Author Rebuttal · Authors · 2025-07-31
>
> We appreciate your questions and concrete suggestions! We respond to specific questions below. We look forward to further discussion, and welcome additional comments.
>
> > **Q1) Stochastic visual response functions**
>
> You are correct. Our current framework assumes a true response function that is noiseless, and is only affected by the current stimulus. Due to this assumption, our encoder outputs are also deterministic in nature. We adopted this setup because these conditions are commonly (sometimes implicitly) assumed in other fMRI encoder works.
>
> If we want to construct an encoder that can capture the inherent distribution of a stochastic visual response function, we need to adopt a stochastic generative framework. There has been recent work on adopting diffusion models as fMRI encoders [1]; however their work currently does not capture the trial variability due to averaging across stimulus repeats. In such a framework, the variability across subjects, machines, stimuli, repeats, etc. can be modeled by introducing additional conditioning variables to the generative framework.
>
> We agree this is an interesting future direction, and stochastic fMRI encoders could allow us to more accurately model the visual response.
>
> > **Q2) Differentiability of the visual response function**
>
> To clarify, we **do not assume that the true biological visual response function is inherently differentiable**. Rather, we employ differentiable functions as computational approximations to model these unknown response patterns. We utilize differentiable functions because they can be optimized using modern gradient-based optimization techniques popularized by deep learning.
>
> Modern neural encoding models in computational neuroscience commonly adopt this approach (such as [1,2]) because differentiable neural networks provide universal approximation capabilities, meaning they can approximate non-differentiable functions to high precision [3,4].
>
> As shown in our **Experiments** section, BraInCoRL demonstrates that this differentiability constraint does not meaningfully limit our model's ability to capture biologically relevant response patterns. The successful identification of known category-selective regions, strong cross-dataset generalization, and attention patterns all indicate that differentiable approximations effectively capture the essential semantic selectivity of higher visual cortex.
>
> > **Q3) Choices of Losses**
>
> This is a good question! Per your suggestion, we conduct an ablation study on the choice of different loss functions used to optimize the network on real neural data.
>
> Our experimental results show that MSE and L1 losses achieve similar performance across all context sizes, with minimal differences. This suggests that both metrics are equally effective for capturing voxelwise neural response patterns. We also explore the hybrid loss of 0.5 × MSE loss + 0.5 × (1 - cosine similarity), where the cosine loss is applied on ground truth voxel weights that are derived from ridge regression. We find this underperforms by 2-4% compared to MSE/L1. We believe this is due to the higher weighting on directional correctness of the weights, rather than optimizing for prediction error minimization.
>
> We are motivated to adopt MSE since it corresponds to OLS regression commonly used for fMRI. MSE loss is also suitable because it provides optimal performance.
>
> * Voxel-wise explained variance of different training losses with the CLIP backbone for Subject 1 (higher is better).
>
> |  # In-Context Samples | 30| 70| 100| 300| 512| 700| 1000|
> |--------------------|---------------|---------------|---------------|---------------|---------------|---------------|---------------|
> | **MSE**| **0.0991** | **0.1283** | **0.1374** | **0.1571** | **0.1628** | **0.1650** | **0.1662** |
> | L1| 0.1005 | 0.1286 | 0.1375 | 0.1568 | 0.1626 | 0.1650 | 0.1650 |
> | MSE + CosSim| 0.0927 | 0.1213 | 0.1285 | 0.1498 | 0.1571 | 0.1593 | 0.1623 |
>
> > **Q4) Clarity of text**
>
> Thank you for pointing out the typos and labels in our paper! We will update the paper according to your suggestions and those from other reviewers to improve the clarity of our upcoming revision.
>
> > **Conclusion**
>
> We are grateful for your suggestions and detailed comments. We will incorporate your feedback and our corrections into the upcoming revision. We look forward to discussing with you further!
>
>
> [1] MindSimulator: Exploring Brain Concept Localization via Synthetic FMRI (2025).
>
> [2] A differentiable approach to multi-scale brain modeling (2024).
>
> [3] Multilayer feedforward networks are universal approximators (1989).
>
> [4] Approximation capabilities of multilayer feedforward networks (1991).

---

> > ### Comment · Reviewer_CYvz · 2025-08-02
> >
> > I thank the authors for answering all of my questions and for the additional assessment of different loss functions. I find all of the answers convincing and clarrifying. I feel reassured in my original assessment.

---

> > > ### Author Response · Authors · 2025-08-04
> > >
> > > We are grateful for your positive evaluation of this paper!
> > >
> > > As a gentle reminder, for NeurIPS this year you may have to edit your review with a "Final Justification". Thank you for taking the time!

---

### Official Review · Reviewer_BDTo · 2025-07-03

**Clarity:** 3
**Significance:** 4
**Originality:** 3
**Rating:** 5
**Confidence:** 4

**Summary:**

This work proposes a meta-learning approach to modeling fMRI recordings in response to visual stimuli. The method allows for better generalization between subjects and even recording conditions, taking a step beyond subject-specific encoding models. A transformer is trained to predict linear-readout weights given a relatively small set of in-context image-feature, voxel-activation pairs. The model is trained on the NSD dataset and tested on held-out subjects, as well as the BOLD5000 dataset. Results show that the proposed method explains more variance than direct optimization using Ridge regression across a variety of in-context training set sizes. Further, the manuscript presents clusterings of predicted predicted readout weights in accordance with known category-selective brain regions. An analysis of the images with highest attention scores in the last transformer layer reveals that images tend to stem from the category preferred by the respective brain region.

**Questions:**

1. A brief overview of the model architecture would be good to include in the main text.

2. In l.131 should it be [x_i;\beta_i]? Or what does v_i denote exactly?

3. Equation (4) came very out of the blue for me – perhaps this could be explained more thoroughly

4. Is it possible to use the full 9,000 images as in-context samples, thereby actually reaching the performance of the fully trained model?

5. The results reported on page 6 are somewhat unclear to me. It says '...we evaluated BrainCoRL with each subject's 9000 unique images as in-context support set...'. Then in the next sentence '...using just 100 test(?) images', In the table caption, it says '...uses just 100 in-context images...'. Perhaps this could be clarified a little.

6. In Fig.4 for BOLD5000, why are so few samples used compared to NSD?

**Ethical Concerns:**

["NO or VERY MINOR ethics concerns only"]

**Final Justification:**

The thorough discussion of the questions I've raised has reinforced my position that this manuscript is suitable for publication.

**Limitations:**

yes

**Paper Formatting Concerns:**

I have no concerns.

**Quality:**

3

**Strengths And Weaknesses:**

**Strengths**

1. The idea of applying meta-learning to fMRI data is interesting and as noted in the manuscript, it is a promising approach towards more subject-general models.

2. The results clearly show data-efficiency and transfer abilities of the proposed model. This has strong relevance for the field.

3. Experiments are quite rigorous, testing multiple models and datasets.

4. Cluster analyses and semantic characterizations of the predicted voxel weights show clear correspondences to established category-selective regions, showing that the model does learn meaningful features.

5. The manuscript is clear and easy to follow for the most part.

**Weaknesses**

1. Fig. 3b) shows that the performance difference between the pretrained model and BrainCoRL is non-zero, but relatively small. A lot of the advantage over Ridge regression is seemingly not attributable to explicit leveraging of the structure of neural data.

2. The variance explained by all models is relatively low, which means the model is competitive, but at a low overall level. We observe that it is able to capture very rough trends like category selectivity, but it is difficult to gauge whether learning more fine-grained features driving single voxels is possible with this method. Since a voxel's tuning properties are inferred by studying the tuning properties of other voxels, I could imagine a smoothing effect where voxels with very distinct tuning properties are not modeled accurately. Therefore, while the feature analyses in Fig. 6-8 are interesting, they are not fully convincing for me. This is a drawback but not a strong criticism for this work – I acknowledge that this can be left for future efforts.

---

> ### Author Rebuttal · Authors · 2025-07-31
>
> We appreciate the concrete feedback and comments! We address your suggestions below.
>
> > **Q1) Performance gain from real neural data**
>
> This is a good question. To clarify, `Figure 3b` evaluates performance at a fixed in-context support size of ***100 images on a held out subject***. To further evaluate the full impact of real-data finetuning, we visualize the trends as shown in `Section A.6` on page 30 of the Supplement. We find that finetuning yields substantial performance gains -- over 20% improvement in variance explained across support set sizes compared to the pretrained model. This effect becomes especially pronounced at larger context sizes (>500), where **the synthetic pretrained model often falls below the Ridge regression baseline**, while the **fully trained BraInCoRL model continues to improve**.
>
> Beyond raw performance, we want to highlight how BraInCoRL leverages the structure of neural data. During Stage 3 (real-data finetuning), the model meta-learns across a distribution of voxel-level tasks -- each defined by a voxel’s stimulus-response mapping in a specific subject. BraInCoRL uses a transformer-based hypernetwork conditioned on in-context stimulus-response pairs. This design allows the model to capture shared representational structure across subjects and voxels, and adapt to novel subjects and voxels through in-context learning, without test-time training.
>
> > **Q2) Finer-grained selectivity**
>
> We agree that current models (our model, as well as those from other researchers) based on linear projections of features from a frozen foundation model vision transformer, impose strong constraints on the final fMRI encoding model. This may make the learning of fine-grained features that drive voxel activity difficult. We would like to clarify that our reported explained variance is not normalized to the effective noise ceiling -- and the explained variance of our models are very competitive with recent state-of-the-art work [1] that utilize a vision transformer as the image backbone.
>
> In the future, a promising direction for fMRI encoders may be constructing non-linear mappings between backbone features and the brain, or those that can capture a distribution of activations based on stochastic generative models. We anticipate that a primary driver of this type of work will be the availability of higher quality and more diverse fMRI datasets.
>
> > **Q3) Model architecture**
>
> The architecture of our model is composed of:
>
> (1) An **input projection** MLP layer is applied to each token individually, which maps the stimulus semantics and voxel activation into an embedding.
>
> (2) 20 **Transformer encoder** layers that integrate information across all in-context examples and computes trends for voxel activation across multiple stimuli.
>
> (3) The `[CLS]` token from the final layer goes through an MLP to output a hyperweight which is used to parameterize the final encoder.
>
> Our code is provided on `line 22` of the main paper. We will update the paper revision to include additional model details.
>
> > **Q4) Typo clarification**
>
> In line 131, we correct the context token $c_i$ to be $=[x_i; \beta_i]$, so $\beta$ will be consistent in the previous BOLD signal representation in line 129.
>
> > **Q5) Logit scaling equation clarification**
>
> Here we clarify the use of logit-scaling in `Equation (4)` for our model.
>
> Logit scaling is motivated by the need for robust generalization across varying context sizes. Specifically, it helps the in-context encoder maintain stable performance regardless of the number of conditioning stimuli, including those beyond the training range. Although we train with between $30$ and $500$ images, practical applications may involve fewer than $30$ or more than $500$. Logit scaling ensures the model remains functional and accurate even in these extrapolated settings.
>
> This logit scaling approach was originally proposed in [2, 3], and later adopted by Qwen LLM (logn-scaling) [4] & Llama 4 LLM (temperature scaling) [5]. Below, we provide a brief summary of the high-level mathematical rationale based on [2], along with our own commentary:
>
> The self-attention value for query token $i$ to value token $j$ is
>
> $$a\_{i,j} = \frac{e^{\lambda \mathbf{q}\_i \cdot \mathbf{k}\_j}}{\sum\_{j=1}^n e^{\lambda \mathbf{q}\_i \cdot \mathbf{k}\_j}}$$
>
> Then we define the entropy of query token $i$ as $$\mathcal{H}\_i = -\sum\_{j=1}^n a\_{i,j} \log a\_{i,j}$$
>
> If we substitute the expression for $a\_{i,j}$, we will have
>
> $$\mathcal{H}\_i = -\sum\_{j=1}^n a\_{i,j} \log \left(\frac{e^{\lambda \mathbf{q}\_i \cdot \mathbf{k}\_j}}{\sum\_{j=1}^n e^{\lambda \mathbf{q}\_i \cdot \mathbf{k}\_j}}\right)$$
> Since $\sum\_{j=1}^n a\_{i,j} = 1$,
>
> we can express the entropy as
>
> $$\mathcal{H}\_i = \log \sum\_{j=1}^n e^{\lambda \mathbf{q}\_i\cdot \mathbf{k}\_j} - \frac{\sum\limits\_{j=1}^n e^{\lambda \mathbf{q}\_i\cdot \mathbf{k}\_j}(\lambda \mathbf{q}\_i\cdot \mathbf{k}\_j)}{\sum\limits\_{j=1}^n e^{\lambda \mathbf{q}\_i\cdot \mathbf{k}\_j}}$$
>
> If the first term is expressed as an expectation over $j$, we have
>
> $$\sum\_{j=1}^n e^{\lambda \mathbf{q}\_i\cdot \mathbf{k}\_j} = n\times \frac{1}{n}\sum\_{j=1}^n e^{\lambda \mathbf{q}\_i\cdot \mathbf{k}\_j}\approx n\mathbb{E}\_j[e^{\lambda \mathbf{q}\_i\cdot \mathbf{k}\_j}]$$
>
> Then we can derive the following approximation of entropy
> $$\mathcal{H}\_i \approx \log n + \log \mathbb{E}\_j[e^{\lambda \mathbf{q}\_i\cdot \mathbf{k}\_j}] - \frac{\lambda\mathbb{E}\_j[e^{\lambda \mathbf{q}\_i\cdot \mathbf{k}\_j}(\mathbf{q}\_i\cdot \mathbf{k}\_j)]}{\mathbb{E}\_j[e^{\lambda \mathbf{q}\_i\cdot \mathbf{k}\_j}]}$$
>
> Assuming that the vectors are the output of a layernorm layer with length $\sqrt{d}$, the expectation can be converted to one over the angles between vectors
>
> $$\mathcal{H}\_i \approx \log n + \log \mathbb{E}\_{\theta}[e^{\lambda d \cos\theta}] - \frac{\lambda d\,\mathbb{E}\_{\theta}[e^{\lambda d \cos\theta}\cos\theta]}{\mathbb{E}\_{\theta}[e^{\lambda d \cos\theta}]}$$
>
> Since randomly distributed vectors in high-dimensional spaces are mostly nearly orthogonal, we derive a term that can be approximately expressed as
>
> $$\mathcal{H}\_i \approx \log n + C$$
> where $C$ does not depend on the number of tokens $n$.
>
> This yields an approximate scaling factor for the logits of $\log n$, which helps maintain the entropy of the attention distribution across different context lengths.
>
> As a result, we apply logit scaling with a factor of $\log n$ to the $QK^T$ term in the attention mechanism, changing the standard attention value formulation.
>
> Note that in the explanation we adopt the notation from [2].
>
> **We have provided a more detailed discussion of the use of Equation (4) in the revised version of our paper.**
>
> > **Q6) BraInCoRL conditioned on 9000 images**
>
> This is a great suggestion. We have added an additional experiment to evaluate the performance of BraInCoRL with 9000 in-context samples using the CLIP backbone, for the four NSD subjects (S1, S2, S5, S7) in our main paper. Note that in this case, the model has not seen the subject we test on during training, and the model was trained on the other three subjects.
>
> As shown in the table, the performance difference is less than $0.01$ explained variance difference across all subjects when compared to a fully trained model (which is fit on each subject's entire 9000-image training set). This means BraInCoRL achieves 94-99% of the fully trained model's performance. Crucially, BraInCoRL reaches this near-optimal performance while requiring no parameter updates or gradient-based optimization during the in-context inference.
>
> * Voxel-wise explained variance of BraInCoRL with the CLIP backbone compared with the fully trained reference model. The difference variance explained is less than 1% (higher is better).
>
> |Subject|S1|S2|S5|S7
> |-|-|-|-|-
> |Fully Trained|0.1765|0.1882|0.2310|0.1554
> |BraInCoRL|0.1667|0.1817|0.2225|0.1541
>
> > **Q7) Number of images clarification**
>
> In the result on page 6, we intend to say that we can **utilize up to** each subject's 9000 unique images in the NSD dataset as the in-context support set. But during the evaluation experiment (such as shown in Table 1 caption), we just randomly sample a subset of e.g. 100 in-context images as the inference input. **We have clarified this in the revision.**
>
> > **Q8) Size of BOLD5000**
>
> This discrepancy arises from the inherent nature of the BOLD5000 dataset itself. The NSD dataset involves up to $10,000$ images for each subject being viewed up to three times. In contrast, for BOLD5000, only approximately $100$ images were viewed more than three times.
>
> We were advised by the BOLD5000 authors to only analyze the image subset with more than three repeats. Following our discussions with the BOLD5000 dataset authors, we follow the sampling strategy below: we randomly sample 20% of the BOLD5000 images as inference images, with the remaining 80% used as the in-context support set.
> > **Conclusion**
>
> We are very grateful for the highly detailed and actionable suggestions.
> We have included additional details for our method, and conducted additional experiments. We look forward to any additional questions.
>
> [1] Better models of human high-level visual cortex emerge from natural language supervision with a large and diverse dataset (2023).
>
> [2] Analyzing the scale operation of attention from the perspective of entropy invariance (2021). (see line 564 of main paper).
>
> [3] Overcoming a Theoretical Limitation of Self-Attention (2022).
>
> [4] Qwen technical report (2023).
>
> [5] The Llama 4 herd: The beginning of a new era of natively multimodal ai innovation (2025).

---

> > ### Comment · Reviewer_BDTo · 2025-08-01
> >
> > Thank you for the detailed clarifications and comments. I do believe that this is a strong contribution and recommend it be accepted for publication.
> >
> > The only question that remains for me to some degree is that in Q2): You noted that _all_ models based on linear projections of features from a frozen foundation model vision transformer impose strong constraints on the final fMRI encoding model (which I am certainly not trying to argue against). However, my question was geared more towards your approach specifically: Since the weights for a test voxel are predicted based on information learned from _other_ voxels, do you foresee a problem with performance on voxels with very distinct tuning patterns that might be learnable using Ridge (since Ridge doesn't care about how 'unusual' the weights are w.r.t. other voxels)?

---

> ### Author Response · Authors · 2025-08-01
>
> This is an interesting question. We agree this is a possibility.
>
> To expand on this, our work and ridge regression utilize different inductive biases. Ridge regression penalizes strong contributions from individual features and is equivalent to a zero-mean gaussian prior with variance determined by the penalty term. On the other-hand, our framework learns a data-driven inductive bias over many voxel/stimuli pairings. As stated by the "No Free Lunch" theorem, there is no universally better learning algorithm -- so there will be cases where ridge may outperform our method when the visual response falls significantly outside our training distribution.
>
> However, fMRI's inherent spatial structure favors our approach. Each voxel aggregates signals from ~100,000 neurons, inherently smoothing population-level neural responses. This spatial integration:
> 1. Suppresses single-neuron variability
> 2. Creates low-dimensional representations of neural population activity (think of the extreme case where a voxel is an average of all neurons in the brain)
> 3. Creates regularities that data-driven methods can utilize
>
> Ridge regression’s static bias (weight shrinkage) cannot explicitly leverage this structure, while our data-driven method can.

---

> > ### Comment · Reviewer_BDTo · 2025-08-04
> >
> > Thank you for your response - this is an interesting discussion of the question and the argument about the inherent structure of fMRI data is very convincing.

---

> > > ### Author Response · Authors · 2025-08-04
> > >
> > > We are very grateful for the positive assessment you've given to our work!
> > >
> > > We would like to again express our appreciation for your suggestions.

---

### Official Review · Reviewer_yVtB · 2025-07-04

**Clarity:** 3
**Significance:** 3
**Originality:** 3
**Rating:** 5
**Confidence:** 5

**Summary:**

This paper introduces BraInCoRL, a novel transformer-based model for brain encoding that performs meta-learning and in-context learning to predict voxel-level responses in the human higher visual cortex to visual stimuli. Unlike traditional brain encoders that require subject-specific large-scale data, BraInCoRL is trained across subjects and generalizes to novel subjects and datasets using only a small number of in-context examples (i.e., image–response pairs).

**Questions:**

Complexity and Scalability of the Training Pipeline

The proposed three-stage training strategy—synthetic pretraining, context extension, and real-data finetuning—is well-motivated and effective. However, it introduces significant complexity and resource requirements. In practical settings, especially those with limited access to large-scale fMRI datasets or compute, this may hinder broader adoption. Additionally, the contribution of each stage to the final model performance is not thoroughly disentangled.

Suggestion for Authors
It would strengthen the paper to include a quantitative evaluation of each stage’s contribution to performance. For instance, how much does synthetic pretraining alone contribute to generalization? How critical is the context extension step? Ablations isolating each component, or alternative streamlined training setups, would help clarify the necessity of the full pipeline and guide future, potentially more efficient implementations.

Overall comments:

This paper presents a compelling advance in the intersection of computational neuroscience and meta-learning. The proposed BraInCoRL model is technically innovative, empirically strong, and provides new insights into how generalizable brain encoding models can be built.

**Ethical Concerns:**

["NO or VERY MINOR ethics concerns only"]

**Final Justification:**

The rebuttal addressed my questions.

**Limitations:**

Yes.

**Paper Formatting Concerns:**

No concerns.

**Quality:**

3

**Strengths And Weaknesses:**

Strong Generalization Across Subjects and Datasets
BraInCoRL shows consistent improvements across novel subjects and even across datasets collected with different scanners and protocols (e.g., NSD to BOLD5000). It operates effectively with just a few in-context images, outperforming ridge regression baselines and closely matching fully trained models using 9,000 images.

Modular and Versatile Model Design
The model is constructed modularly with a frozen CLIP feature extractor and a higher visual cortex transformer that can generate hyperweights for voxel-wise encoding. This enables efficient processing of both visual and language inputs, facilitating downstream tasks such as natural language–driven cortical querying.

Insightful Analyses on Model Behavior
The paper presents compelling insights into the behavior of the model:
It learns to attend to semantically relevant images when predicting voxel responses in category-selective regions.

Surprisingly, it performs better when in-context support sets are randomly sampled rather than overly specific, suggesting that semantic diversity enhances generalization—a notable finding for the design of few-shot neural systems.

---

> ### Author Rebuttal · Authors · 2025-07-31
>
> We appreciate your excellent suggestions! We will incorporate all of your feedback into our paper.
>
> > **Q1) Training cost and complexity**
>
> We agree with you that the training stage of our model is more costly than traditional fMRI encoders. However this cost gains us significant generalization ability.
>
> Our current model already achieves strong generalization performance on fMRI data that was collected using a different scanner, voxel size, experiment protocol, and human subjects -- **without requiring any finetuning or training on these smaller datasets**. We also achieve significant data-efficiency gains, demonstrating better performance in the low data regime compared to traditional baselines.
>
> Due to this, our **training & inference code and weights will be made available under a permissive open source license**. Future researchers will be able to adopt our model for their datasets without training from scratch (and so will not be limited to their small datasets), and benefit from the models that we have already trained.
>
>
> > **Q2) Contribution of each stage**
>
> The three-stage training is essential, with each stage addressing a distinct challenge.
>
> * **1st stage: Synthetic Pretraining** enables comprehensive coverage of the visual response space. It provides the model with an initial and broad feature representation by exposing it to a large volume of synthetically generated data.
> * **2nd stage: Context Extension** trains the model on variable-length contexts, enabling robust length generalization, and allows for test-time context size to be outside of the range seen during training.
> * **3rd stage: Real-Data Finetuning** adapts the model to the statistical properties of real fMRI signals, in order to bridge the gap between synthetic and biological domains. Note that in the main paper, for evaluation fairness we avoid training on the same subject that we evaluate on.
>
> This progression from synthetic foundation → context flexibility → biological adaptation ensures that each fundamental challenge—response function coverage, variable context handling, and biological realism—is systematically addressed in the optimal order.
>
> Due to OpenReview limitations, here we present results only for the first two subjects of the NSD dataset; however, the observed trends and performance are consistent across all other subjects.
>
> * Voxel-wise explained variance for BraInCoRL with CLIP backbone for NSD Subject 1 on different training stages, compared with ridge baseline and fully trained reference model (higher is better).
>
> | # In-Context Samples | 30  | 70  | 100  | 300  | 512  | 700  | 1000  |
> |--------|-----|-----|-----|-----|-----|-----|-----|
> | Fully Trained (9000 images) | 0.1765 | 0.1765 | 0.1765 | 0.1765 | 0.1765 | 0.1765 | 0.1765 |
> | Synthetic Pretraining (stage 1) | 0.0508 | 0.0886 | 0.1034 | 0.1183 | 0.1177 | 0.1153 | 0.1086 |
> | Context Extension (stage 2) | 0.0461  | 0.0906 | 0.1054 | 0.1314 | 0.1381 | 0.1396 | 0.1415 |
> | **Real-Data Finetuning (stage 3)** | **0.0991**  | **0.1283** | **0.1374** | **0.1571** | **0.1628** | **0.1650** | **0.1662** |
> | Ridge Baseline | 0.0227 | 0.0548 | 0.0697 | 0.1081 | 0.1237 | 0.1311 | 0.1402 |
>
> * Voxel-wise explained variance for BraInCoRL with CLIP backbone for NSD Subject 2 on different training stages, compared with ridge baseline and fully trained reference model (higher is better).
>
> | # In-Context Samples | 30  | 70  | 100  | 300  | 512  | 700  | 1000  |
> |--------|-----|-----|-----|-----|-----|-----|-----|
> | Fully Trained (9000 images) | 0.1882 | 0.1882 | 0.1882 | 0.1882 | 0.1882 | 0.1882 | 0.1882 |
> | Synthetic Pretraining (stage 1) | 0.0579 | 0.0968 | 0.1121 | 0.1354 | 0.1338 | 0.1317 | 0.1257 |
> | Context Extension (stage 2) | 0.0632 | 0.1038 | 0.1157 | 0.1402 | 0.1467 | 0.1502 | 0.1526 |
> | **Real-Data Finetuning (stage 3)** | **0.1128**  | **0.1412** | **0.1516** | **0.1720** | **0.1780** | **0.1796** | **0.1813** |
> | Ridge Baseline | 0.0262 | 0.0624 | 0.0787 | 0.1211 | 0.1358 | 0.1453 | 0.1538 |
>
> In stage 1, we find that training with synthetic data without context extension limits the performance at higher context sizes. After stage 2, the model gain the ability to consistently improve performance as context size increases. In stage 3, after being exposed to real neural data, our model further improves in performance.
>
> > **Conclusion**
>
> We thank you for your comments, and we hope that this clarifies our results! We have further clarified the contributions of each training stage, and will update the paper to reflect your suggestions.

---

> ### Author Response · Authors · 2025-08-06
> **Gentle reminder**
>
> Hello,
>
> We hope our response has been helpful! We would be happy to discuss further.
>
> As a gentle reminder, due to NeurIPS requirements this year -- you may have to submit a "Mandatory Acknowledgement" and edit your review with a "Final Justification".
>
> Thank you!

---

### Official Review · Reviewer_H8UU · 2025-07-05

**Clarity:** 3
**Significance:** 3
**Originality:** 3
**Rating:** 5
**Confidence:** 3

**Summary:**

This paper addresses an important problem: how to construct cross-subject, few-shot fMRI encoding models. From my understanding, the paper appears to use a small set of image- (single) voxel pairs as a support set, and based on this, trains a model to output the parameters of voxel-wise encoding model. However, I did not fully grasp some details of the proposed method. The paper evaluates its approach on datasets such as NSD, and the experimental results show that the proposed method can effectively encode fMRI signals.

**Questions:**

For the methodology section, I have the following unclear points:
+ In line 131, the context tokens c is indexed by i, which seems to imply that its range is from 1 to p, however, in line 132, its range is from 1 to k. it seems inconsistent?
+ The authors use beta to represent the voxel's BOLD signal (such as in line 129), but in the context tokens c, they directly use the "voxel" v. Does this v here refer to a vector similar to a positional encoding?
+ The authors employ the technique of logit scaling in Section 3.3; however, I am not familiar with it. Could the authors provide a clearer explanation to clarify its underlying principle?

For the experimental section, I have the following questions:
+ What is explained variance? Why is only this metric used in the evaluation on the NSD dataset, while the Pearson correlation coefficient is not assessed?
+ Are there additional metrics to directly assess the accuracy of fMRI encoding?

**Ethical Concerns:**

["NO or VERY MINOR ethics concerns only"]

**Final Justification:**

After carefully reading the authors' response, all of my questions have been fully addressed. This paper is highly insightful for the research field, and therefore I have raised my rating.

**Quality:**

3

**Strengths And Weaknesses:**

Strengths:
+ The paper addresses an important research problem and achieves promising results.
+ The paper provides a relatively comprehensive review of related work.
+ The meta-learning method proposed in the paper is thought-provoking and offers valuable insights.

Weaknesses (Details refer to Questions):
+ In my opinion, the presentation of the method should be more detailed to improve clarity and comprehensibility.

---

> ### Author Rebuttal · Authors · 2025-07-30
>
> We appreciate your very thoughtful and helpful suggestions. Below are our responses. We look forward to further discussion, and welcome additional comments.
>
> > **Q1) Notation regarding token indexing**
>
> The context tokens in `line 129` and `line 132` should be indexed by the same variable and extend to the same count. **This is a typo that we have updated in our upcoming revision.** In both cases, the number of context tokens during inference represents the number of stimuli that a subject has seen -- which are then made available to the model for in-context learning.
>
> **We will update this in the revised paper.**
>
> > **Q2) Notation regarding voxel signal**
>
> We agree that the clarity here could be improved. Per your suggestion, we have unified the notation around voxel $v$ and beta value $\beta$. Our original intention was to use $\beta$ since the voxel's response to a given image in fMRI viewing is typically derived as the beta coefficients of a GLM regression. We intend the two variables to represent the same value, which is a scalar representing a voxel's response to a given stimulus.
>
> **This has been changed in our revision.**
>
> > **Q3) Principles underlying logit scaling**
>
> The motivating factor underlying logit scaling is our desire to have our in-context learned encoder perform well *regardless of the number of stimuli given to the model*, and effectively generalize to context sizes beyond those seen during training. For example, while we may only train with between $30$ and $500$ images, a third-party experimenter may want to use fewer than $30$ images or more than $500$ images to condition the model. Across all cases, we want the model to succeed.
>
> This logit scaling method was first proposed in [1, 2], and later adapted in Qwen LLM (logn-scaling) [3] & Llama 4 LLM (temperature scaling) [4]. We will briefly summarize the high-level math, which we take from [1] with our commentary:
>
> Let the attention value in self-attention for a particular query token $i$ to value token $j$ to be
>
> $$a\_{i,j} = \frac{e^{\lambda \mathbf{q}\_i \cdot \mathbf{k}\_j}}{\sum\_{j=1}^n e^{\lambda \mathbf{q}\_i \cdot \mathbf{k}\_j}}$$
>
> Then the entropy is defined as $$\mathcal{H}\_i = -\sum\_{j=1}^n a\_{i,j} \log a\_{i,j}$$
>
> Subsituting the expression for $a\_{i,j}$ we have the entropy as
>
> $$\mathcal{H}\_i = -\sum\_{j=1}^n a\_{i,j} \log \left(\frac{e^{\lambda \mathbf{q}\_i \cdot \mathbf{k}\_j}}{\sum\_{j=1}^n e^{\lambda \mathbf{q}\_i \cdot \mathbf{k}\_j}}\right)$$
> Since $\sum\_{j=1}^n a\_{i,j} = 1$,
>
> we can express the formula as
>
> $$\mathcal{H}\_i = \log \sum\_{j=1}^n e^{\lambda \mathbf{q}\_i\cdot \mathbf{k}\_j} - \frac{\sum\limits\_{j=1}^n e^{\lambda \mathbf{q}\_i\cdot \mathbf{k}\_j}(\lambda \mathbf{q}\_i\cdot \mathbf{k}\_j)}{\sum\limits\_{j=1}^n e^{\lambda \mathbf{q}\_i\cdot \mathbf{k}\_j}}$$
>
> If the first term is expressed as an expectation over $j$, we have
>
> $$\sum\_{j=1}^n e^{\lambda \mathbf{q}\_i\cdot \mathbf{k}\_j} = n\times \frac{1}{n}\sum\_{j=1}^n e^{\lambda \mathbf{q}\_i\cdot \mathbf{k}\_j}\approx n\mathbb{E}\_j[e^{\lambda \mathbf{q}\_i\cdot \mathbf{k}\_j}]$$
>
> Which leads to the following approximation of entropy
>
> $$\mathcal{H}\_i \approx \log n + \log \mathbb{E}\_j[e^{\lambda \mathbf{q}\_i\cdot \mathbf{k}\_j}] - \frac{\lambda\mathbb{E}\_j[e^{\lambda \mathbf{q}\_i\cdot \mathbf{k}\_j}(\mathbf{q}\_i\cdot \mathbf{k}\_j)]}{\mathbb{E}\_j[e^{\lambda \mathbf{q}\_i\cdot \mathbf{k}\_j}]}$$
>
> If the vectors are assumed to be the output of a layernorm layer with length $\sqrt{d}$, the expectation can be converted to one over the angles between vectors
>
> $$\mathcal{H}\_i \approx \log n + \log \mathbb{E}\_{\theta}[e^{\lambda d \cos\theta}] - \frac{\lambda d\mathbb{E}\_{\theta}[e^{\lambda d \cos\theta}\cos\theta]}{\mathbb{E}\_{\theta}[e^{\lambda d \cos\theta}]}$$
>
> Since most randomly distributed vectors in higher dimensions are orthogonal, we derive a term which can roughly be expressed as
>
> $$\mathcal{H}\_i \approx \log n + C$$
> where $C$ does not depend on the number of tokens $n$.
>
> This leads to an approximate scaling factor for the logits of $\log n$ to keep the entropy invariant to context length.
>
> Therefore, we change the standard formulation of attention values by applying a scaling factor of $\log n$ to the $QK^T$ term, as shown in `Equation (4)` of our paper.
>
> Note that in the above explanation we adopt the notation from [1].
>
> **We have provided a more detailed discussion of the use of logit scaling in the revised version of our paper.**
>
> > **Q4) Why explained variance?**
>
> Explained Variance is a performance measuring metric that quantifies the proportion of the total variance in the fMRI signal that can be predicted by our BraInCoRL encoding model.
>
> We follow prior work that utilizes the NSD fMRI dataset, which primarily evaluate ***Explained Variance*** as the metric of encoding model performance [5, 6, 7] -- note all three of these papers have one NSD author as co-author.
>
> We agree that Pearson $R$ is also a possible metric we can use to evaluate encoder performance. In a sense, Pearson $R$ is ***more forgiving metric***. Pearson $R$ is affected neither by scaling, nor by constant bias, while explained variance is only robust to bias. The use of Pearson $R$ reflects an implicit preference for the model to capture the right trend in predicted responses while caring less about capturing the exact value.
>
> Note that BOLD5000 contains distribution shifts for several reasons - for example, it incorporates COCO, ImageNet, and SUN images. With NSD, we explicitly controlled the training and evaluation responses distributions to be the same, making explained variance appropriate as a strong criterion. In contrast, with BOLD5000, even though every voxel was still z-scored, there are still many factors that are different compared to NSD (MRI scanner quality, more variable task). After discussion with BOLD5000 authors, we were advised to evaluate Pearson $R$ for BOLD5000, rather than the more stringent explained variance (see also the GLMSingle paper for a direct comparison of the datasets).
>
> > **Q5) Other metrics for NSD**
>
> Here we evaluate Pearson $R$, and Spearman's rank correlation coefficient (Spearman's $\rho$) for NSD dataset. Note that in this case, the BraInCoRL model has not been trained or finetuned on `Subject 1`, while the `Fully Trained` model is trained on $9000$ images from this subject. Our model can approach within 10% of the `Fully Trained` model using just $300$ images when evaluated using Pearson $R$.
>
>
> * Voxel-wise Pearson R for BraInCoRL, within-subject ridge regression baseline and fully-trained reference model (NSD dataset, CLIP backbone, Subject 1, higher is better).
>
> |# In-Context Samples|30|70|100|300|512|700|1000|
> |-|-|-|-|-|-|-|-|
> |Fully Trained|0.3934|0.3934|0.3934|0.3934|0.3934|0.3934|0.3934|
> |BraInCoRL|0.2899|0.3289|0.3411|0.3679|0.3757|0.3790|0.3811|
> |Ridge Baseline|0.1776|0.2287|0.2547|0.3059|0.3282|0.3384|0.3506|
>
> * Voxel-wise Spearman's $\rho$ for BraInCoRL, within-subject ridge regression baseline and fully-trained reference model (NSD dataset, CLIP backbone, Subject 1, higher is better).
>
> |# In-Context Samples|30|70|100|300|512|700|1000|
> |-|-|-|-|-|-|-|-|
> |Fully Trained|0.3736|0.3736|0.3736|0.3736|0.3736|0.3736|0.3736|
> |BraInCoRL|0.2721|0.3088|0.3215|0.3489|0.3570|0.3604|0.3636|
> |Ridge Baseline|0.1304|0.1969|0.2218|0.2849|0.3070|0.3173|0.3289|
>
> > **Q6) Other metrics for BOLD5000**
>
> Here we evaluate explained variance and Spearman's rank correlation (Spearman's $\rho$) for BOLD5000. Our BraInCoRL model was trained on NSD, while the Ridge Baseline was fit on BOLD5000. We achieve significantly better performance.
>
> * Voxel-wise EV for BraInCoRL and within-subject ridge regression baseline (BOLD5000 dataset, CLIP backbone, Subject CSI1, higher is better).
>
> |# In-Context Samples|10|20|30|40|50|60|70|80|
> |-|-|-|-|-|-|-|-|-|
> |BraInCoRL|0.0114|0.0155|0.0259|0.0355|0.0421|0.0454|0.0504|0.0531|
> |Ridge Baseline|0.0041|0.0079|0.0057|0.0035|0.0093|0.0155|0.0244|0.0266|
>
> * Voxel-wise Spearman's $\rho$ for BraInCoRL and within-subject ridge regression baseline (BOLD5000 dataset, CLIP backbone, Subject CSI1, higher is better).
>
> |# In-Context Samples|10|20|30|40|50|60|70|80|
> |-|-|-|-|-|-|-|-|-|
> |BraInCoRL|0.1130|0.1382|0.1561|0.1694|0.1769|0.1846|0.1878|0.1905|
> |Ridge Baseline|0.0790|0.0916|0.1016|0.1192|0.1317|0.1411|0.1480|0.1508|
>
> >**Conclusion**
>
> Thank you for the detailed and thoughtful feedback. Following your suggestions, we have clarified our motivation with utilizing logit scaling in `section 3.3`, we have provided reasons why explained variance was used for NSD, and we have run additional experiment to evaluate different regression metrics on NSD and BOLD5000. **We will update our revision to incorporate these changes.**
>
> We hope our answers address your questions, and look forward to any additional discussion.
>
>
> [1] Analyzing the scale operation of attention from the perspective of entropy invariance (2021). (see line 564 of main paper).
>
> [2] Overcoming a Theoretical Limitation of Self-Attention (2022).
>
> [3] Qwen technical report (2023).
>
> [4] The Llama 4 herd: The beginning of a new era of natively multimodal ai innovation (2025).
>
> [5] Non-neural factors influencing BOLD response magnitudes within individual subjects (2022).
>
> [6] A 7T fMRI dataset of synthetic images for out-of-distribution modeling of vision (2025).
>
> [7] Better models of human high-level visual cortex emerge from natural language supervision with a large and diverse dataset (2023).

---

### Note · Authors · 2025-08-12

We are grateful to all reviewers for their constructive suggestions, which we agree will significantly improve the communication of our work.

We are very encouraged by reviewers’ evaluation on the quality of this paper. All four reviewers find the work interesting ("**paper is thought-provoking and offers valuable insights**" (H8UU); "**compelling insights into the behavior of the model**" (yVtB); "**a promising approach towards more subject-general models**" (BDTo); "**this study presents a solid modelling framework**" (CYvz)).

### General clarifications
### 1. Scope and experiments
* We introduce a meta-learned in-context model of human higher visual cortex. To our knowledge, we are ***the first*** to construct a predictive model of the brain that can transfer across ***non-overlapping stimuli sets, subjects, scanners, and protocols without requiring any training or finetuning***.
* Our work utilizes meta-learning across response functions, and in-context learning across stimuli/responses pairs. This allows our model to flexibly adapt to any number of stimuli.
* We show that our method can achieve up to ***10x data-savings*** while maintaining similar predictive performance on held-out subjects (Figure 3).
* Our method is a crucial step for building **foundational human neural models** in fMRI and other modalities, which will allow the field to extend modern neuroAI models to a broader range of populations and tasks.

### 2. New results
1. To address the suggestion of reviewer H8UU, we add results using additional regression metrics and find that our method achieves consistently good performance.
2. To address the suggestion of reviewer yVtB, we add results that highlight the contribution of each training stage.
3. For reviewer BDTo, we add a result showing how our method compares when using the full 9000 training set.
4. For reviewer CYvz, we compare different loss functions used during training.

### 3. Writing
We thank all reviewers for suggestions regarding our writing and clarity. We have corrected some typos in our notation, and added theoretical details clarifying why we utilize logit scaling. We believe that the clarifications suggested by the reviewers will improve the communication of our work.

### 4. Conclusion
We genuinely appreciate the suggestions, and believe our paper will be improved with your feedback! The additional experiments and clarifications will be reflected in the final version as well.

Best,

Authors

---

### Decision · Program_Chairs · 2025-09-17

**Decision:**

Accept (poster)

**Comment:**

The paper presents a cross-subject fMRI encoding model based on in-context learning. The method, BraInCoRL, is valided against a simple Ridge regression baseline, but performance and trade-offs of the method are validated in quantitative and qualitative analysis throughout the paper. Post discussion, the reviewers remain unanimously positive about this work and recommend acceptance. I follow the recommendation, provided that the new results from the rebuttal phase are worked into the paper and the supplementary materials.